# PHUMA: Physically-grounded Humanoid Locomotion Dataset

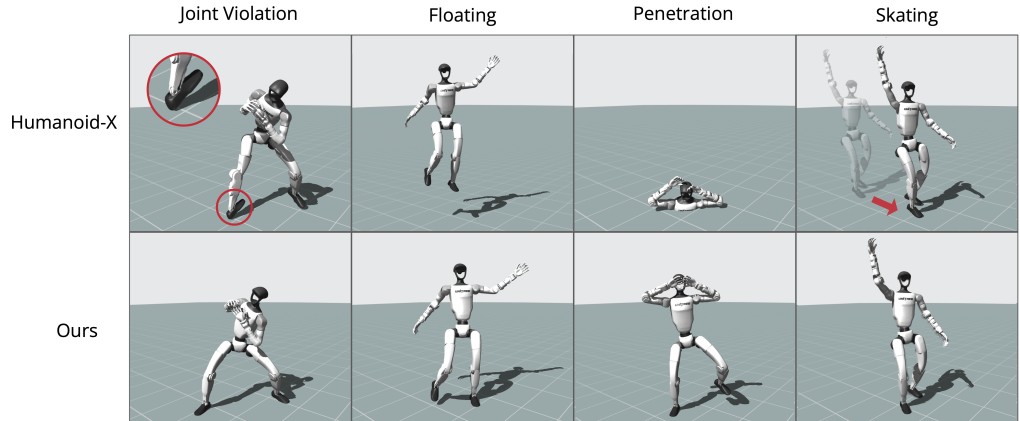

Figure 1: **Physical reliability of Humanoid-X vs. PHUMA**. Each column illustrates four failure modes: joint violation, floating, penetration, and skating. Humanoid-X (Mao et al., 2025) (top row) often exhibits these issues due to direct video-to-motion conversion, while PHUMA (bottom row) mitigates those violations through careful data curation and physically grounded retargeting.

## Abstract

Motion imitation is a promising approach for humanoid locomotion, enabling agents to acquire humanlike behaviors. Existing methods typically rely on high-quality motion capture datasets such as AMASS, but these are scarce and expensive, limiting scalability and diversity. Recent studies attempt to scale data collection by converting large-scale internet videos, exemplified by Humanoid-X. However, they often introduce physical artifacts such as floating, penetration, and foot skating, which hinder stable imitation. In response, we introduce **PHUMA**, a **P**hysically-grounded **HUMA**noid locomotion dataset that leverages human video at scale, while addressing physical artifacts through careful data curation and physics-constrained retargeting. PHUMA enforces joint limits, ensures ground contact, and eliminates foot skating, producing motions that are both large-scale and physically reliable. We evaluated PHUMA in two sets of conditions: (i) imitation of unseen motion from self-recorded test videos and (ii) path following with pelvis-only guidance. In both cases, PHUMA-trained policies outperform Humanoid-X and AMASS, achieving significant gains in imitating diverse motions. Qualitative videos are available at the https://anonymous-robotics-researcher.github.io/Paper_10482.

## 1 Introduction

Humanoid robots are central to the pursuit of general-purpose embodied AI, but their deployment in real-world first requires locomotion that is both stable and humanlike. While reinforcement learning (RL) with task-oriented rewards has led to remarkable progress in quadrupedal locomotion (Hwangbo et al., 2019; Lee et al., 2020; Tan et al., 2018), directly applying these strategies to

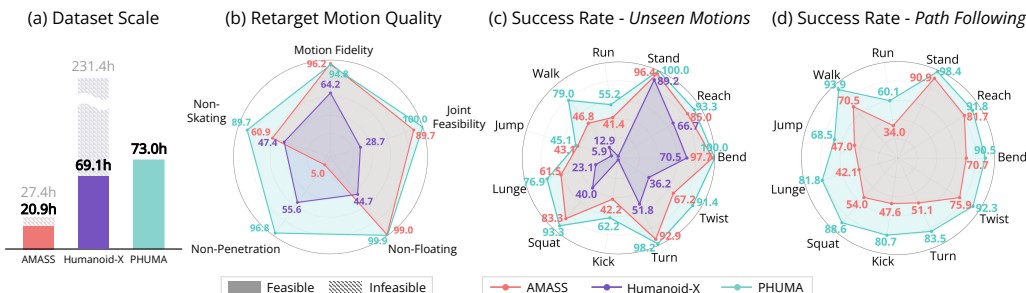

Figure 2: **Overview of datasets and performance.** PHUMA is both large-scale and physically reliable, which translates into higher success rates in motion imitation and pelvis path following. (a) Feasible and infeasible human motion sources in each dataset. (b) Physical reliability, with AMASS retargeted using a standard learning-based inverse kinematics method. (c) Success rate on unseen motions. (d) Success rate in path-following. Results are reported on the Unitree G1 humanoid.

humanoids often produces gaits that are effective yet non-humanlike (Hansen et al., 2023; Sferrazza et al., 2024). To address this limitation, motion imitation has emerged as a promising paradigm. In motion imitation, policies are trained to replicate human movements through a three-stage pipeline: (1) collecting human motion data, (2) retargeting it to the robot's morphology, and (3) using RL to track the retargeted trajectories (Peng et al., 2018; Tessler et al., 2024; He et al., 2024b).

Despite its promise, progress in motion imitation is fundamentally constrained by the scale, diversity, and physical feasibility of human motion data. High-quality motion capture datasets such as LaFAN1 (Harvey et al., 2020) and AMASS (Mahmood et al., 2019) provide a high proportion of physically feasible motions, but are limited in scale and diversity, with motions dominated by simple motions such as reaching and walking. To overcome this scarcity, recent work has sought to scale data collection by leveraging vast internet videos. Humanoid-X (Mao et al., 2025) exemplifies this trend by converting videos to SMPL representations (Loper et al., 2023) using a video-to-motion model (Kocabas et al., 2020), then retargeting them to humanoid embodiments. However, this pipeline introduces two types of physical violations. First, the video-to-motion model often misestimates global pelvis translation, producing artifacts such as floating or ground penetration. Second, the retargeting stage prioritizes joint alignment over physical plausibility (He et al., 2024b;a), leading to joint violation and foot skating as illustrated in the top row of Figure 1.

In response, we introduce **PHUMA**: **P**hysically-grounded **HUMA**noid locomotion dataset that leverages large-scale human video while overcoming physical artifacts through careful data curation and physics-constrained retargeting. As illustrated in Figure 3(1), we first collect diverse high-quality human motion data and filter out infeasible motions from Humanoid-X, such as root jitter or actions requiring external objects like sitting on chairs. This filtering removes approximately 70% of the original dataset, as shown in Figure 2(a). As shown in Figure 3(2), we then apply Physically constrained Shape-adaptive Inverse Kinematics (PhySINK), which enforces soft joint limits, ground contact, and anti-skating constraints to eliminate violations such as joint overextension, floating, and sliding. As a result, PHUMA provides substantially more physically plausible motions than existing datasets, 349.9% more than AMASS and 5.5% more than Humanoid-X (Figure 2(a,b)).

We validate the effectiveness of PHUMA in two settings: (i) imitation of unseen motions and (ii) path following with pelvis-only guidance. Using the MaskedMimic framework Tessler et al. (2024) for RL training, we tested policies on Unitree G1 and H1-2 humanoids. On 504 self-recorded videos across 11 motion types, policies trained with PHUMA achieve 1.2x and 2.1x higher success rates than AMASS and Humanoid-X, respectively (Figure 2(c)). For path following, PHUMA-trained policies improve overall success rate by 1.4x over AMASS, with 1.6x gains in vertical (e.g., squat, lunge, jump) and 2.1x gains in horizontal (e.g., walk, run) motion path trajectories(Figure 2(d)). We will release PHUMA as a public resource to advance future research in humanoid locomotion.

## 2 RELATED WORK

PHUMA focuses on constructing a large-scale, physically reliable humanoid locomotion dataset, requiring two components: (1) collection of diverse human motion data and (2) retargeting of these motion data to the humanoid robot.

### 2.1 HUMAN MOTION DATA

Human motion data, typically represented in the SMPL format (Loper et al., 2023; Pavlakos et al., 2019), is obtained from two main sources: motion capture systems and reconstruction from video (Gu et al., 2025). Motion capture data (CMU, 2003; Zhang et al., 2022; Al-Hafez et al., 2023) provides accurate kinematics but is difficult to scale due to its reliance on complex instrumentation, such as multi-camera arrays and marker-based suits. Even a relatively large dataset like LaFAN1 (Harvey et al., 2020) contains only a few hours of motion. AMASS (Mahmood et al., 2019), the most extensive and widely-used dataset, remains dominated by walking motions in indoor labs.

Recent datasets (Lin et al., 2023; Zhang et al., 2025; Chung et al., 2021; Cai et al., 2022; Tsuchida et al., 2019) leverage the scalability and diversity of human videos. Humanoid-X (Mao et al., 2025) is notable for massively scaling up from Internet video data, providing an abundant collection of data from motion capture and video recovery. However, video-derived motion often exhibits severe jitter across frames (Kocabas et al., 2020; Wang et al., 2024), and motion from either source is susceptible to physical artifacts such as interactions with unmodeled objects (e.g., sitting on a chair that does not exist) (Luo et al., 2023; 2024) and implausible foot-ground contact, including floating or penetration (Goel et al., 2023; Ye et al., 2023; Yu et al., 2021; Ugrinovic et al., 2024).

To mitigate these issues, recent works have introduced automated data cleaning strategies (Luo et al., 2024). ASAP (He et al., 2025a) employs a "sim-to-data" cleaning procedure that uses a motion tracking policy in a physics simulator to filter out failures. However, this process primarily relies on the tracking policy rather than the physical validity of the motion, which may be biased by the tracking performance of the policy. KungfuBot (Xie et al., 2025) instead adopts physics-based principles for data cleaning, such as stability criteria, and utilizes contact masks for motion correction. However, they rely on zero-velocity assumptions and ankle height for contact estimation, which are often unreliable for video-reconstructed motions.

PHUMA is a large-scale, diverse, and curated motion dataset aggregated from both motion capture and human video through a physics-aware curation pipeline, which corrects implausible foot-ground contact and filters out corrupted sequences with severe physical artifacts.

### 2.2 HUMANOID MOTION RETARGETING

Human motion data, widely used for physics-based character control (Peng et al., 2018; Wagener et al., 2022; Luo et al., 2021; 2023; Hansen et al., 2025; Tessler et al., 2024; Tirinzoni et al., 2025), is now also being applied to the field of humanoid robotics (Radosavovic et al., 2024a; Fu et al., 2024; Cheng et al., 2024; Ji et al., 2024; Chen et al., 2025; Xie et al., 2025; Truong et al., 2025; Li et al., 2025). For instance, Humanoid Policy ~ Human Policy (Qiu et al., 2025) leverages egocentric human video for manipulation, while ASAP (He et al., 2025a) utilizes retargeted human motion to learn agile locomotion. This relies on motion retargeting, which is critical for adapting human movements to humanoid robots that, despite their morphological similarities to humans, possess distinct kinematic and proportional characteristics (Kim et al., 2025; Ho et al., 2010; Zhang et al., 2023).

A primary challenge is motion mismatch, where the retargeted motion fails to capture the kinematic pose of the source. Inverse kinematics (IK) methods (Radosavovic et al., 2024b; Zakka, 2025; Ze et al., 2025) often overlook the differences in body shape, resulting in unnatural motions like in-toed gaits. Recently, GMR (Araujo et al., 2025) demonstrated that IK can yield highly plausible results through careful engineering, yet it remains reliant on heuristic scale adjustments and is prone to artifacts such as foot floating. Shape-adaptive inverse kinematics (SINK) methods, introduced by H2O (He et al., 2024b), address this by first adapting the source human model to match the body shape and limb proportions of the target robot. The motion is then aligned to the source by matching global joint positions (He et al., 2024a; 2025b) or local limb orientations (Cheynel et al., 2023;

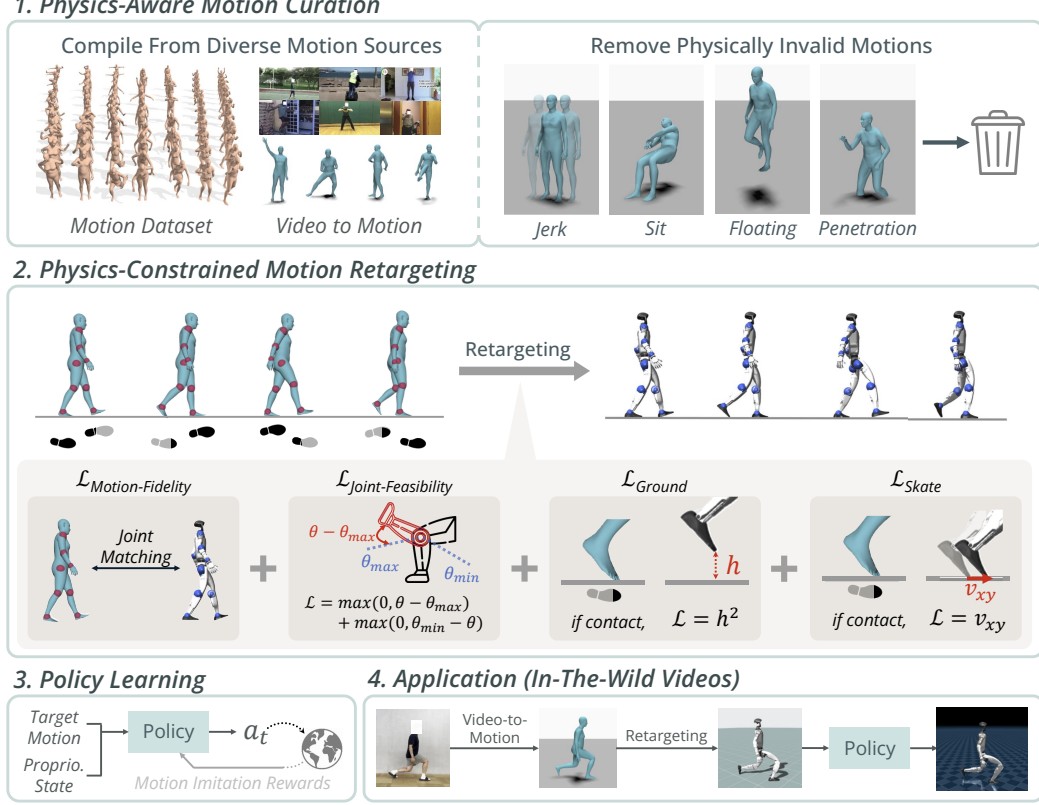

Figure 3: **Overview of the PHUMA pipeline.** Our four-stage pipeline for motion imitation learning includes: (1) Motion Curation, where we filter out problematic motions from a diverse dataset; (2) Motion Retargeting, where the filtered motions are retargeted to the humanoid using PhySINK, incorporating a series of losses.; (3) Policy Learning, where a policy is trained to imitate the retargeted motions; and (4) Inference, where the trained policy is used to control the humanoid, enabling it to imitate motions from unseen videos processed by a video-to-motion model.

Allshire et al., 2025). While effective at pose matching, SINK approaches are physically under-constrained, introducing artifacts including joint limit violations and implausible ground interactions such as floating, penetration, and skating.

Physically constrained shape-adaptive inverse kinematics (PhySINK) directly addresses these physical artifacts by augmenting the optimization with joint feasibility, grounding, and skating loss terms, ensuring the retargeted motion maintains fidelity to the source while remaining physically plausible.

## 3 METHOD

Our goal is to construct PHUMA, a large-scale, physically reliable dataset for humanoid locomotion. We build upon the Humanoid-X motions (Mao et al., 2025), which are rich in scale but exhibits physical artifacts. We first apply physics-aware curation to filter out problematic motions (Section 3.1). Next, to solve artifacts introduced during the retargeting process itself, we employ PhySINK, our physics-constrained retargeting method that adapts the curated motion to the humanoid while enforcing physical plausibility (Section 3.2). Our two-stage pipeline is illustrated in Figure 3.

### 3.1 PHYSICS-AWARE MOTION CURATION

The goal of our curation pipeline is to refine raw motion data, which often contains artifacts that make the motion physically implausible for a humanoid. Our process targets key issues such as severe jitter, instabilities from interactions with unmodeled objects, and incorrect foot-ground contact.

Table 1: **Composition of the PHUMA dataset.** A summary of the number of clips and duration for each sub-dataset, categorized by source and scene. PHUMA aggregates these diverse sub-datasets, resulting over 73 hours of physically-grounded motion clips.

| Dataset | # Clip | # Frame | Duration | Source | Scene |
|---|---|---|---|---|---|
| LocoMuJoCo (Al-Hafez et al., 2023) | 0.78K | 0.93M | 0.86h | Motion Capture | Indoor |
| GRAB (Taheri et al., 2020) | 1.73K | 0.20M | 1.88h | Motion Capture | Indoor |
| EgoBody (Zhang et al., 2022) | 2.12K | 0.24M | 2.19h | Motion Capture | Indoor |
| LAFAN1 (Harvey et al., 2020) | 2.18K | 0.26M | 2.40h | Motion Capture | Indoor |
| AMASS (Mahmood et al., 2019) | 21.73K | 2.25M | 20.86h | Motion Capture | Indoor |
| HAA500 (Chung et al., 2021) | 1.76K | 0.11M | 1.01h | Human Video | Outdoor |
| Motion-X Video (Lin et al., 2023) | 33.04K | 3.45M | 31.98h | Human Video | Outdoor |
| HuMMan (Cai et al., 2022) | 0.50K | 0.05M | 0.47h | Human Video | Indoor |
| AIST (Tsuchida et al., 2019) | 1.75K | 0.18M | 1.66h | Human Video | Indoor |
| IDEA400 (Lin et al., 2023) | 9.94K | 0.98M | 9.10h | Human Video | Indoor |
| **PHUMA Video** | **0.50K** | **0.06M** | **0.56h** | Human Video | Outdoor |
| **PHUMA** | **76.01K** | **7.88M** | **72.96h** | | |

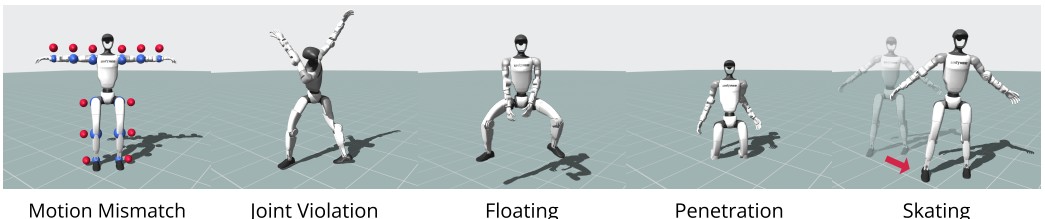

Figure 4: **Common physical artifacts in motion retargeting.** From left to right: Motion Mismatch, Joint Violation, Floating, Penetration, and Skating.

To mitigate high-frequency jitter, we apply a low-pass Butterworth filter (Appendix A.1.1). We identify unstable motions, such as sitting on a non-existent chair, by calculating the center-of-mass (CoM) distance from the base of support. To correct foot-ground contact, a consistent ground plane in the world frame is essential. Since recovered motions are often defined in a camera's coordinate frame, they lack a true ground reference, which causes floating and penetration. We establish a global ground plane using a majority-voting scheme: each foot vertex contributes to identifying the most consistent contact height. The entire motion is then shifted to align this plane at a height of zero (Appendix A.1.2), after which we compute per-region foot contact scores.

With a reliable ground plane established, we segment all sequences into 4-second clips. We then discard any clip exhibiting: (i) excessive jerk, (ii) a CoM position far outside its support base, or (iii) insufficient foot-ground contact. This chunk-and-filter process maximizes the retention of viable segments from longer, partially flawed sequences (Appendix A.1.3). Finally, we augment these curated motions with data from LaFAN1, LocoMuJoCo, and our own video captures.

As detailed in Table 1, the resulting PHUMA dataset is a large-scale collection containing 73.0 hours of physically plausible motion across 76.0K clips.

## 3.2 PHYSICS-CONSTRAINED MOTION RETARGETING

Inverse kinematics (**IK**) methods often fail to preserve motion style, while shape-adaptive inverse kinematics (**SINK**) preserves style but introduces artifacts such as joint violations and unrealistic ground interactions (Figure 4). Our method, physically constrained shape-adaptive inverse kinematics (PhySINK), overcomes these issues by extending SINK with joint feasibility, grounding, and anti-skating losses, producing motions that are both stylistically faithful and physically plausible.

**Motion Fidelity Loss.** We optimize the humanoid joint positions $q_t$ and root translation $\gamma_t$ over time $t$, so that the retargeted motion closely matches the human motion. The $\mathcal{L}_{\text{Fidelity}}$ is defined as:

$$\mathcal{L}_{\text{global-match}} = \sum_t \sum_i \left\| p_i^{\text{SMPL-X}}(t) - p_i^{\text{Humanoid}}(t) \right\|_1 \tag{1}$$

Table 2: **Quantitative comparison and ablation study of retargeting methods.** We evaluate performance on two humanoids, G1 and H1-2, showing the progressive impact of adding each of our proposed physical constraint losses.

| | Motion Fidelity (%) | Joint Feasibility (%) | Non-Floating (%) | Non-Penetration (%) | Non-Skating (%) |
|---|---|---|---|---|---|
| **(a) G1** | | | | | |
| IK | 27.6 | 91.7 | 55.6 | 47.8 | 59.7 |
| GMR | 56.3 | 81.8 | 14.7 | 100.0 | 67.7 |
| SINK | 94.8 | 95.9 | 96.4 | 14.9 | 55.4 |
| + Joint Feasibility Loss | **94.9** | **100.0** | 96.4 | 14.8 | 55.6 |
| + Grounding Loss | **94.9** | **100.0** | 99.9 | **97.2** | 53.6 |
| + Skating Loss = **PhySINK** | 94.8 | **100.0** | 99.9 | 96.8 | **89.7** |
| **(b) H1-2** | | | | | |
| IK | 36.3 | 80.9 | 57.7 | 45.2 | 56.1 |
| SINK | 93.9 | 15.3 | 42.2 | 81.4 | 47.9 |
| + Joint Feasibility Loss | **94.0** | **99.9** | 44.4 | 79.9 | 50.7 |
| + Grounding Loss | 93.9 | **99.9** | 99.8 | **98.1** | 49.3 |
| + Skating Loss = **PhySINK** | 93.9 | **99.9** | 99.7 | 97.7 | **87.7** |

$$\mathcal{L}_{\text{local-match}} = \sum_t \sum_{i \neq j} m_{ij} \underbrace{\left\| \Delta p_{ij}^{\text{SMPL-X}}(t) - \Delta p_{ij}^{\text{Humanoid}}(t) \right\|_2^2}_{\text{position}} \tag{2}$$

$$+ \sum_t \sum_{i \neq j} m_{ij} \underbrace{\left( 1 - \left\langle \Delta p_{ij}^{\text{SMPL-X}}(t), \ \Delta p_{ij}^{\text{Humanoid}}(t) \right\rangle \right)}_{\text{orientation}}$$

$$\mathcal{L}_{\text{smooth}} = \sum_t \left\| \dot{q}_t - 2\dot{q}_{t+1} + \dot{q}_{t+2} \right\|_1 + \sum_t \left\| \dot{\gamma}_t - 2\dot{\gamma}_{t+1} + \dot{\gamma}_{t+2} \right\|_1 \tag{3}$$

$$\mathcal{L}_{\text{Fidelity}} = w_{\text{global-match}} L_{\text{global-match}} + w_{\text{local-match}} L_{\text{local-match}} + w_{\text{smooth}} L_{\text{smooth}} \tag{4}$$

where $p_i^{\text{SMPL-X}}(t)$ and $p_i^{\text{Humanoid}}(t)$ denote the global 3D position of joint $i$ at time $t$. $\Delta p_{ij}$ denotes the position difference between joints $i$ and $j$. $m_{ij}$ is a binary mask that equals 1 when $i$ and $j$ are immediate neighbors in the humanoid kinematic tree, and 0 otherwise. We define *Motion Fidelity (%)* as the average percentage of frames where the mean per-joint position error is below 10 cm and the mean per-link orientation error is below 10 degrees.

**Joint Feasibility Loss.** Configurations that violate joint limits can lead to unrealistic motion or instabilities in a simulator. $L_{\text{Joint Feasibility}}$ penalizes joint angles and velocities that approach or exceed the predefined operational limits of the humanoid:

$$\mathcal{L}_{\text{position-violation}} = \sum_t \left[ \max(0, q_t - 0.98 q_{\max}) + \max(0, 0.98 q_{\min} - q_t) \right] \tag{5}$$

$$\mathcal{L}_{\text{velocity-violation}} = \sum_t \left[ \max(0, \dot{q}_t - 0.98 \dot{q}_{\max}) + \max(0, 0.98 \dot{q}_{\min} - \dot{q}_t) \right] \tag{6}$$

$$\mathcal{L}_{\text{Feasiblity}} = \mathcal{L}_{\text{position-violation}} + \mathcal{L}_{\text{velocity-violation}}. \tag{7}$$

We define *Joint Feasibility (%)* as the percentage of frames where all joint positions and velocities remain within 98% of their predefined mechanical limits.

**Grounding Loss.** The grounding loss corrects for floating or penetration artifacts by enforcing that the foot regions of the humanoid remain on the ground plane during frames with detected contact:

$$\mathcal{L}_{\text{Ground}} = \sum_{i \in \{\text{LH,LT,RH,RT}\}} \sum_t c_t^i \left\| p_t^i(z) \right\|_2^2 \tag{8}$$

where $c_t$ is a contact score for foot regions Left Heel (LH), Left Toe (LT), Right Heel (RH), and Right Toe (RT) at frame $t$. We define *Non-Floating (%)* as the percentage of contact frames where the foot is within 1 cm above the ground, and *Non-Penetration (%)* as the percentage of contact frames where the foot is within 1 cm below the ground.

**Skating Loss.** The skating loss prevents foot sliding by penalizing the horizontal velocity of any foot region that is in contact with the ground:

$$\mathcal{L}_{\text{Skate}} = \sum_{i \in \{\text{LH,LT,RH,RT}\}} \sum_t c_t^i \left\| \dot{p}_t^i(x, y) \right\|_2 \tag{9}$$

where $c_t$ is a contact score for foot regions Left Heel (LH), Left Toe (LT), Right Heel (RH), and Right Toe (RT) at frame $t$. We define *Non-Skating (%)* as the percentage of contact frames where the foot's horizontal velocity is below 10 cm/s. The objective for the baseline SINK method consists solely of the motion fidelity loss.

Our PhySINK objective is a weighted sum of the motion fidelity loss and the physical constraint terms. By optimizing this augmented objective, PhySINK generates motions that maintain kinematic similarity to the source while being physically plausible.

$$\mathcal{L}_{\text{PhySINK}} = \mathcal{L}_{\text{Fidelity}} + w_{\text{Feasibility}}\mathcal{L}_{\text{Feasibility}} + w_{\text{Ground}}\mathcal{L}_{\text{Ground}} + w_{\text{Skate}}\mathcal{L}_{\text{Skate}} \tag{10}$$

To evaluate PhySINK, we retarget PHUMA to two Unitree robots, G1 (Unitree Robotics, 2025a) and H1-2 (Unitree Robotics, 2025b), and compare against a standard IK solver (Zakka, 2025), and GMR (Araujo et al., 2025), and SINK framework. As shown in Table 2, both the standard IK solver and GMR struggle with motion fidelity. These methods use simple linear scaling to adjust human keypoint positions to the robot's body proportions, which changes the target motion and fails to preserve the original kinematics. This approach alters the target motion without preserving the original human motion's kinematics. In contrast, SINK first optimizes the human body shape to fit the humanoid's proportions, then applies the original joint angles, thereby improving motion fidelity. Building upon SINK, our proposed losses progressively enhance performance: the joint feasibility loss raises feasibility to nearly 100%, and the grounding loss reduces floating and penetration to over 96%. While GMR achieves the best non-penetration performance by optimizing the motion's height based on its minimum ground contact point, it suffers from significant floating artifacts. The full PhySINK model, which incorporates the skating loss, preserves motion fidelity while achieving strong results across all physical metrics, including nearly 90% non-skating performance. Qualitative comparisons between retargeting methods are shown in Figures 8 and 9.

## 4 EXPERIMENTS

In this section, we evaluate the effectiveness of PhySINK and PHUMA along three axes, addressing the following research questions:

**RQ1.** What does our proposed PhySINK retargeting method compare with established retargeting approaches (IK, SINK) in terms of motion imitation performance?

**RQ2.** How effective is PHUMA as a training corpus for motion imitation, compared to prior datasets utilized for humanoid motion (LaFAN1, AMASS, Humanoid-X)?

**RQ3.** When using a simplified controller that considers only pelvis tracking rather than full-body state tracking, does training on PHUMA achieve better path-following performance than training on existing benchmark datasets across various motion categories?

RQ4. Does training the policy with PHUMA lead to improved Sim-to-Sim transfer performance compared to AMASS?

### 4.1 EXPERIMENT SETUP

**Training.** We employ the MaskedMimic framework (Tessler et al., 2024) for all policy training, which provides a unified approach for motion tracking with either full body state or partial body state information (e.g., pelvis-only). The framework trains policies using PPO (Schulman et al., 2017) to imitate human motion by maximizing reward signals that measure tracking accuracy.

For RQ1 and RQ2, we train full-state motion tracking policies. These policies receive current proprioceptive state ($s_t^p$), which includes joint positions, orientations, and velocities, as well as full goal states ($s_t^g$) representing the target motion trajectories. Given these inputs, the policy outputs joint angle commands ($a_t$) that are executed via PD controllers. The reward function is designed to measure how well the humanoid matches the target motion.

For RQ3, we employ the partial-state protocol from MaskedMimic. This involves first training a full-state teacher policy on full-body reference motion data, then using knowledge distillation to train a student policy that mimics the action of the teacher policy while receiving only pelvis position and rotation as input, enabling pelvis path-following control while maintaining humanlike movement.

All experiments are conducted in the IsaacGym simulator using Unitree G1 (29 DoF) and H1-2 (21 DoF, excluding wrist joints). Detailed hyperparameters are provided in Appendix 12, with complete observation space and reward function specifications in Table 10 and Appendix B.2, respectively.

**Evaluation.** To assess the trained policies, we evaluate performance on two distinct datasets. The first consists of about 7.5K motions (10% of PHUMA) that were held out during training. The second comprises 504 self-collected video sequences converted to motion sequences using a video-to-motion model. Processing details for the self-collected videos are provided in Appendix C.1.

For evaluating the full body motion tracking (RQ1, RQ2), we adopt the success rate metric from prior motion imitation studies (He et al., 2024b; 2025a; Xie et al., 2025), which measures the ratio of motions successfully imitated within a specified deviation threshold. Unlike prior work that uses a 0.5m threshold, we employ a stricter 0.15m threshold, as the standard threshold incorrectly classifies scenarios as successful when humanoids remain stationary during jumps or stay upright during squatting motions. Further discussions related to the threshold selection is detailed in Appendix C.2.

In path following settings (RQ3), we use a similar success rate metric focused on pelvis tracking accuracy. Specifically, we measure the ratio of motions where the policy successfully tracks pelvis trajectories within the same 0.15m threshold throughout the motion sequence. To evaluate performance across diverse motion types, we organize all evaluations into four motion categories: stationary (stand, reach), angular (bend, twist, turn, kick), vertical (squat, lunge, jump), and horizontal (walk, run). This categorization allows us to assess how well policies generalize across different types of human locomotion and movement patterns.

## 4.2 PHYSINK RETARGETING METHOD EFFECTIVENESS

To evaluate the effectiveness of our proposed PhySINK retargeting method, we compare it against two established approaches: IK, SINK. We retarget the same source motions from AMASS using all three methods, then train separate full-state motion tracking policies on each retargeted dataset.

Table 3 demonstrates that PhySINK consistently outperforms both baseline methods across all motion categories and humanoid embodiments. The results validate that physically constrained retargeting directly translates to better imitation performance, with improvements particularly pronounced in dynamic motions (vertical and horizontal categories) where physical constraints are most critical.

## 4.3 PHUMA DATASET EFFECTIVENESS

Having demonstrated PhySINK's effectiveness, we now compare PHUMA against existing humanoid datasets. We train full-state policies on four datasets with different characteristics: LaFAN1(small-scale, high-quality), AMASS(medium-scale, moderate-quality), Humanoid-X(large-scale, lower-quality), and PHUMA(large-scale, high-quality). For AMASS, we apply the widely-used SINK retargeting method since it provides human motion source data, while LaFAN1 and Humanoid-X are used directly as pre-existing humanoid datasets.

As shown in Table 4, PHUMA trained policies achieve the highest success rates across all motion categories and both humanoids. The results reveal that neighter scale nor quality alone is sufficient. Humanoid-X, despite its large size, underperforms due to quality issues, while LaFAN1 and AMASS, though cleaner, lack coverage in several motion types. By combining large scale with high quality motions, PHUMA delivers consistently superior performance across diverse behaviors.

## 4.4 PELVIS-ONLY PATH FOLLOWING CONTROL PERFORMANCE

We evaluate whether training on PHUMA enables better pelvis path-following control compared to the AMASS dataset. Using MaskedMimic's partially-constrained protocol, we train two student policies: one distilled from an AMASS-trained teacher and another from a PHUMA-trained teacher. Both students receive only pelvis position and rotation as input.

As shown in Table 5, policies trained on PHUMA consistently outperform those trained on baseline datasets across all motion categories and humanoids. This improvement is particularly pronounced for vertical and horizontal motions, where AMASS shows significant limitations due to its composition of predominantly simpler motions like reaching and turning (Figure **??**). More specifically,

Table 3: **Motion tracking performance across retargeting approaches.** We evaluate the motion tracking success rate of policies trained on AMASS data retargeted by three different methods (IK, SINK, and PhySINK). Performance is assessed across various motion categories using two humanoid robots, G1 and H1-2, and two test sets: PHUMA Test and Unseen Video.

| Retarget | PHUMA Test | | | | | Unseen Video | | | | |
|---|---|---|---|---|---|---|---|---|---|---|
| | Total | Stationary | Angular | Vertical | Horizontal | Total | Stationary | Angular | Vertical | Horizontal |
| **(a) G1** | | | | | | | | | | |
| IK | 52.8 | 75.3 | 43.9 | 24.3 | 44.2 | 54.0 | 80.3 | 54.6 | 32.7 | 43.3 |
| SINK | 76.2 | 88.5 | 72.1 | 56.8 | 66.8 | 70.2 | 90.7 | 75.0 | 62.7 | 44.1 |
| **PhySINK** | **79.5** | **89.9** | **76.1** | **61.1** | **69.5** | **72.8** | **93.3** | **78.2** | **65.5** | **47.3** |
| **(b) H1-2** | | | | | | | | | | |
| IK | 45.3 | 70.9 | 35.7 | 15.2 | 35.0 | 54.2 | 78.0 | 60.7 | 30.1 | 28.6 |
| SINK | 54.4 | 74.9 | 45.9 | 17.2 | 49.6 | 64.3 | 87.3 | 59.7 | 46.0 | 63.9 |
| **PhySINK** | **64.3** | **83.6** | **57.0** | **27.7** | **55.9** | **72.4** | **99.2** | **66.3** | **57.4** | **63.1** |

Table 4: **Motion tracking performance across datasets.** Success rates of policies trained on LaFAN1, AMASS, Humanoid-X, and PHUMA, evaluated across motion categories on humanoid robots G1 and H1-2 using two test sets: PHUMA Test and Unseen Video.

| Dataset | Hours | PHUMA Test | | | | | Unseen Video | | | | |
|---|---|---|---|---|---|---|---|---|---|---|---|
| | | Total | Stationary | Angular | Vertical | Horizontal | Total | Stationary | Angular | Vertical | Horizontal |
| **(a) G1** | | | | | | | | | | | |
| LaFAN1 | 2.4 | 46.1 | 66.1 | 36.2 | 24.0 | 42.5 | 28.4 | 46.9 | 28.4 | 19.6 | 10.5 |
| AMASS | 20.9 | 76.2 | 88.5 | 72.1 | 56.8 | 66.8 | 70.2 | 90.7 | 75.0 | 62.7 | 44.1 |
| Humanoid-X | 231.4 | 50.6 | 78.4 | 43.0 | 26.0 | 31.8 | 39.1 | 78.0 | 39.6 | 23.0 | 6.5 |
| **PHUMA** | 73.0 | **92.7** | **95.6** | **91.7** | **86.0** | **85.6** | **82.9** | **96.7** | **88.0** | **71.8** | **67.1** |
| **(b) H1-2** | | | | | | | | | | | |
| LaFAN1 | 2.4 | 62.0 | 79.3 | 54.7 | 26.6 | 58.9 | 70.8 | 92.4 | 66.7 | 56.4 | 68.2 |
| AMASS | 20.9 | 54.4 | 74.9 | 45.9 | 17.2 | 49.6 | 64.3 | 87.3 | 59.7 | 46.0 | 63.9 |
| Humanoid-X | 231.4 | 49.7 | 74.6 | 40.4 | 17.0 | 37.3 | 60.5 | 88.3 | 60.0 | 48.7 | 39.7 |
| **PHUMA** | 73.0 | **82.7** | **91.5** | **79.5** | **68.1** | **68.4** | **78.6** | **97.5** | **76.8** | **74.5** | **63.8** |

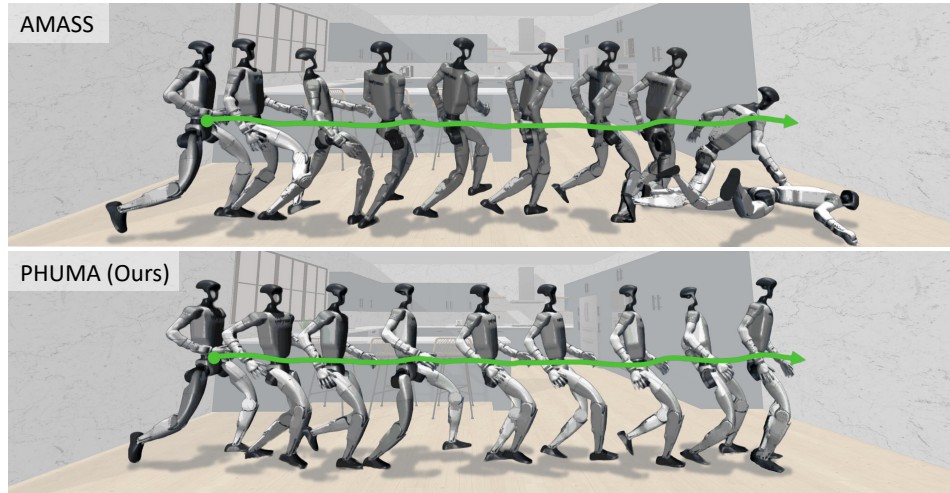

Figure 5: **Path following on running motion.** We visualize the robot's trajectory in a running motion. The target pelvis path is visualized with a green line. Top row presents results from a policy trained on AMASS, while bottom row presents results from a policy trained on PHUMA.

despite AMASS containing numerous walking motions, a substantial performance gap remains in horizontal motions due to the absence of more dynamic movements such as running, as illustrated in Figure 2(d). This limitation is clearly demonstrated in Figure 5, where AMASS-trained policies frequently fail during running motions while PHUMA-trained policies maintain robust performance. These results confirm that PHUMA enables more diverse and dynamic humanoid control compared to AMASS, validating the practical value of PHUMA for complex control.

Table 5: **Pelvis path following peformance across motion dataset.** We evaluate the success rate of pelvis path-following control for policies trained on the AMASS and PHUMA datasets across various pelvis trajectories from the PHUMA Test and Unseen Video.

| Dataset | PHUMA Test | | | | | Unseen Video | | | | |
|---|---|---|---|---|---|---|---|---|---|---|
| | Total | Stationary | Angular | Vertical | Horizontal | Total | Stationary | Angular | Vertical | Horizontal |
| **(a) G1** | | | | | | | | | | |
| AMASS | 60.5 | 85.6 | 60.1 | 51.4 | 66.5 | 54.8 | 83.6 | 66.5 | 33.0 | 27.5 |
| **PHUMA** | **84.5** | **94.6** | **86.1** | **83.7** | **90.2** | **74.6** | **98.3** | **83.3** | **54.3** | **57.1** |
| **(a) H1-2** | | | | | | | | | | |
| AMASS | 60.4 | 84.0 | 62.8 | 43.6 | 78.7 | 72.3 | **96.6** | 77.3 | 52.1 | 72.5 |
| **PHUMA** | **73.9** | **91.2** | **76.5** | **66.9** | **84.8** | **78.1** | **96.6** | **77.8** | **60.6** | **78.0** |

Table 6: **Sim-to-Sim transfer performance across motion dataset.** We evaluate the zero-shot motion tracking success rate of policies trained on PHUMA and AMASS when transferred from the source simulator (Isaac Gym) to the target simulator (MuJoCo). The results are demonstrated on the G1 humanoid to assess robustness against domain shifts in physics engines.

| Dataset | PHUMA Test | | | | | Unseen Video | | | | |
|---|---|---|---|---|---|---|---|---|---|---|
| | Total | Stationary | Angular | Vertical | Horizontal | Total | Stationary | Angular | Vertical | Horizontal |
| AMASS | 62.1 | 81.4 | 54.2 | 38.8 | 64.3 | 64.3 | 86.2 | 68.5 | 54.3 | 37.4 |
| **PHUMA** | **75.0** | **87.6** | **69.3** | **61.6** | **76.3** | **70.0** | **87.9** | **78.8** | **59.6** | **38.5** |

## 4.5 SIM-TO-SIM TRANSFER PERFORMANCE

To validate the environmental robustness of the learned policies, we conduct a Sim-to-Sim transfer experiment. We adopt the training protocol from Kungfubot2 (Han et al., 2025), which has demonstrated strong sim-to-real transfer, while replacing the policy architecture with an MLP. Using this setup, we train motion tracking policies in Isaac Gym on both PHUMA and AMASS, and directly deploy them into the MuJoCo simulator without any fine-tuning. Following the evaluation metrics defined in Kungfubot2 (Han et al., 2025), Table 6 shows that the PHUMA-trained policy consistently outperforms the AMASS baseline across all motion categories in the target simulator. This demonstrates that PHUMA's superior data quality and scale contribute to improved robustness against cross-simulator domain shifts. Detailed performance analysis of both teacher and student policies in the training simulator (Isaac Gym) is provided in Appendix C.3.

## 5 CONCLUSION

We introduced PHUMA, a large-scale, physically plausible humanoid locomotion dataset that overcomes the limitations of existing motion imitation pipelines. Unlike prior video-driven datasets prone to artifacts such as floating, ground penetration, and joint violations, PHUMA combines large-scale human video with careful filtering and our physics-constrained retargeting method, PhySINK, to produce motions that are both diverse and physically reliable. Policies trained on PHUMA consistently outperform those trained on AMASS and Humanoid-X in motion imitation and pelvis-guided path following on Unitree G1 and H1-2 humanoids, demonstrating that progress in humanoid locomotion requires not only scale but also physically reliable data.

Looking forward, future work includes sim-to-real transfer, enabling policies trained with PHUMA to produce physically reliable motions on real humanoid robots, and vision-based control, where video observations replace privileged state inputs to better align with real-world perception.

## REPRODUCIBILITY STATEMENT

To ensure the reproducibility of our results, we provide comprehensive implementation details and experimental specifications. The complete hyperparameter settings for PPO training are detailed in Appendix 12. Our physics-aware curation process and PhySINK retargeting method are described in detail in Sections 3.1 and 3.2, respectively, with algorithmic specifications provided in the appendix. The PHUMA dataset composition and statistics are thoroughly documented in Section 3.1 and Appendix A.2. All evaluation metrics, including our modified success rate threshold and motion category definitions, are explicitly defined in Section 4.1. Implementation details for baseline methods (IK, SINK) follow established protocols as referenced in the main text. The self-collected video processing pipeline is described in Appendix C.1. We plan to release our code, dataset, and trained models upon publication to facilitate further research in this area.

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

# APPENDIX

# A DETAILS OF PHUMA DATASET

## A.1 DATA PREPROCESSING

Before applying inverse kinematics, it is essential to ensure that the human motion data is clean and robust, as this data serves as the target for the humanoid robot to follow. Raw motion data often contains noise from sensor errors, tracking inaccuracies, or estimation artifacts that can negatively impact the retargeting process. To address these issues, we implement the following preprocessing to filter and clean the motion data.

### A.1.1 LOW-PASS NOISE FILTERING FOR MOTION DATA

All motion sequences were resampled to 30 Hz. We smooth all motion channels with a zero-phase, 4th-order Butterworth low-pass filter. For root translation the cutoff is 3 Hz; for global orientation and body pose it is 6 Hz.

### A.1.2 EXTRACTING GROUND CONTACT INFORMATION

We identify a subset of SMPL-X foot vertices that are most indicative of ground interaction. Specifically, we select the 22 vertically lowest vertices from each foot region (left heel, left toe, right heel, right toe) in the SMPL-X default pose, totaling 88 vertices. These vertices are illustrated in Figure 6. The vertex indices corresponding to these ground-contact points are provided in Table 7.

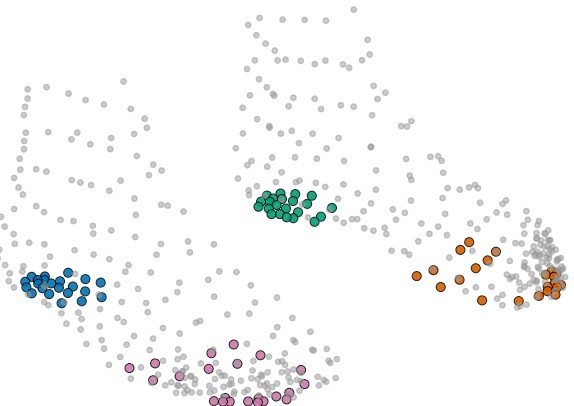

Figure 6: **SMPL-X Foot Vertices for Ground-Contact Detection**. This figure illustrates the selected foot vertices on the SMPL-X model used to detect ground contact. Green and orange points denote the left heel and left toe, while blue and pink represent the right heel and right toe, respectively. The remaining foot vertices are shown in light-gray. The clusters of colored points correspond to the specific parts of the foot that are used to check for contact with the ground, making the process more accurate and robust than using a single point.

Table 7: SMPL-X foot vertex indices used for ground–contact detection.

| Region | Vertex indices |
| --- | --- |
| Left heel | 8888, 8889, 8891, 8909, 8910, 8911, 8913, 8914, 8915, 8916, 8917, 8918, 8919, 8920, 8921, 8922, 8923, 8924, 8925, 8929, 8930, 8934 |
| Left toe | 5773, 5781, 5782, 5791, 5793, 5805, 5808, 5816, 5817, 5830, 5831, 5859, 5860, 5906, 5907, 5908, 5909, 5912, 5914, 5915, 5916, 5917 |
| Right heel | 8676, 8677, 8679, 8697, 8698, 8699, 8701, 8702, 8703, 8704, 8705, 8706, 8707, 8708, 8709, 8710, 8711, 8712, 8713, 8714, 8715, 8716 |
| Right toe | 8467, 8475, 8476, 8485, 8487, 8499, 8502, 8510, 8511, 8524, 8525, 8553, 8554, 8600, 8601, 8602, 8603, 8606, 8608, 8609, 8610, 8611 |

Table 8: Physics-aware data filtering metrics and thresholds.

| Metric | Threshold |
|---|---|
| Root jerk | $< 50 \text{ m/s}^3$ |
| Foot contact score | $> 0.6$ |
| Minimum pelvis height | $> 0.6 \text{ m}$ |
| Maximum pelvis height | $< 1.5 \text{ m}$ |
| Pelvis distance to base of support | $< 6 \text{ cm}$ |
| Spine1 distance to base of support | $< 11 \text{ cm}$ |

To correctly place a motion, it is necessary to establish a single, consistent ground plane. Simple heuristics often fail; defining the ground by the lowest foot position in the sequence can cause floating, while per-frame adjustments introduce jitter. Our method solves this using a majority vote to find the ground height that maximizes the duration of foot contact. In this scheme, each vertex on the feet votes for a potential ground level. The height that gathers the most votes across the entire sequence is selected, as this plane consistently has the most foot vertices near it. The entire motion is then shifted to place this new ground at height zero.

Specifically, we first generate candidate ground coordinates. For each frame $t$, we find the minimum vertical position among these 88 points and record it as a candidate coordinate for the ground plane, $g_t$. Second, we evaluate each candidate $g_t$ by counting the total number of foot vertices, across all frames, that fall within its $\delta = 2.5 \text{ cm}$ tolerance band. We select the candidate $g^\star$ with the highest count as the optimal ground plane and translate the entire sequence vertically to place $g^\star$ at the origin.

### A.1.3 FILTERING MOTION DATA BY PHYSICAL INFORMATION

We evaluate each segmented motion sub-clips based on the metrics summarized in Table 8. Motion sub-clips failing to satisfy these thresholds are discarded.

**Root jerk** represents rapid changes in root acceleration, indicative of abrupt or unnatural motions. High root jerk segments are excluded to ensure smooth and physically plausible trajectories.

**Foot contact score** measures the consistency and sufficiency of foot-ground interactions based on graded ground-contact signals defined by vertex proximity to the ground. Specifically, given a sub-clip with $T$ frames, the foot contact score is computed as:

$$\text{Foot contact score} = \frac{1}{T} \sum_{t=1}^{T} \max \left( c_t^{lh}, c_t^{lt}, c_t^{rh}, c_t^{rt} \right), \tag{11}$$

where $c_t^{lh}, c_t^{lt}, c_t^{rh}$, and $c_t^{rt}$ represent the graded ground-contact ratio at frame $t$ for the left heel, left toe, right heel, and right toe, respectively. A low foot contact score indicates significant penetration or floating, both of which are undesirable artifacts. Note that motions involving airborne phases, such as jumps, can easily satisfy this criterion as long as contact before and after the airborne phase is consistent.

**Pelvis height** criteria exclude segments where the humanoid is unnaturally positioned. Specifically, the minimum height criterion filters out motions that involve the humanoid being excessively crouched or lying on the ground, while the maximum height criterion eliminates segments exhibiting unnatural floating.

**Distance to the base of support** criteria ensure stable and physically plausible balance. Since the SMPL-X model's center of mass typically lies between the pelvis and spine1 joints, deviations of these joints' horizontal-plane projections from the base of support indicate imbalance or instability infeasible for humanoids. The base of support is defined as the convex hull formed by the horizontal-plane projections of the left foot, right foot, left ankle, and right ankle joints.

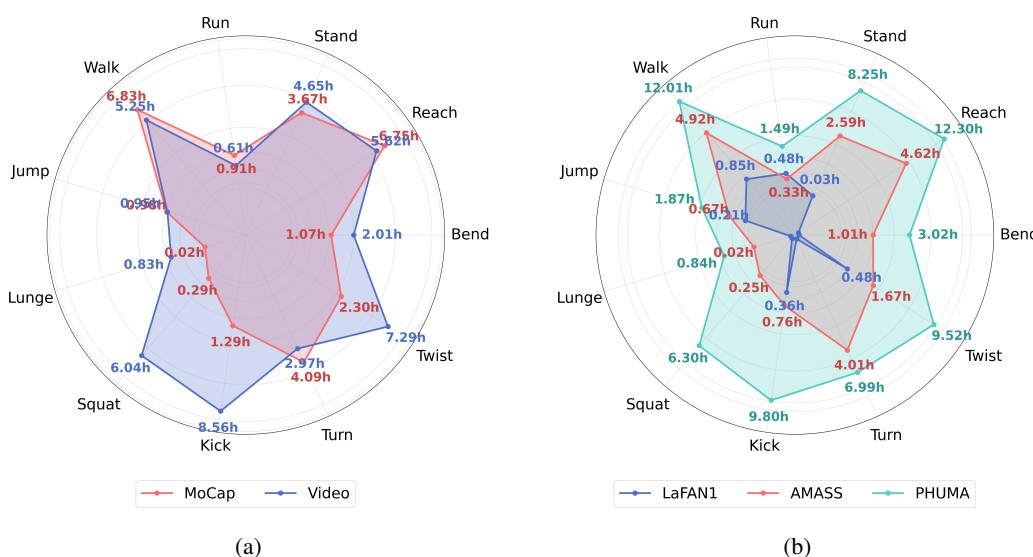

Figure 7: **Comparison of Motion Distributions.** (a) Comparison between MoCap and Video sources in PHUMA. (b) Comparison with other existing datasets.

## A.2 DATASET COMPOSITION AND STATISTICS

This section presents the detailed motion statistics of PHUMA. As we collect the motion data from diverse sources, from MoCap data to video, PHUMAresults in a well-balanced motion distribution that avoids domination by specific motion types. Figure 7b demonstrates that PHUMA exhibits significantly more balanced motion coverage compared to existing datasets. While LaFAN1 and AMASS show uneven distributions with some motion types having very limited motions, lacking certain motion categories entirely (such as reach, bend, and squat motions), or being heavily dominated by specific motions (reach, turn, and walk), PHUMA provides more balanced coverage across all motion categories with substantially more examples per motion type.

This improved diversity and scale directly translate to better imitation performance. Table 4 demonstrates that a policy trained on PHUMA achieves superior overall performance on unseen motions compared to policies trained on other datasets. The results also show consistent performance improvements across all individual motion categories. The results confirm that the enhanced dataset composition benefits generalization across all diverse movement types, indicating that the balanced motion distribution of PHUMA leads to more robust imitation policies.

## A.3 QUALITATIVE COMPARISON OF RETARGETING METHODS

To provide an intuitive comparison of different retargeting approaches, we present qualitative results in Figure 8. Using a walking motion as an example, we demonstrate the distinct characteristics and limitations of each method.

Traditional inverse kinematics (IK) prioritizes matching end-effector positions, such as hands and feet, from rigidly scaled human motions. However, this approach produces unnatural locomotion patterns where the humanoid appears to walk on a tightrope rather than exhibiting a natural human-like gait. This occurs because the fixed scaling cannot account for the proportional differences between human and robot morphologies.

Learning-based inverse kinematics (SINK) generates more natural-looking walking motions compared to traditional IK by optimizing body proportions. However, SINK suffers from physical violations that compromise motion realism. Common issues include foot penetration through the ground surface and fixed ankle angles that result from the lack of explicit contact constraints during the retargeting process.

Table 9: **Imitation Performance: GMR vs. PhySINK Retargeting on Unitree G1.** We evaluate the imitation performance of the MaskedMimic policy trained using datasets retargeted with GMR and PhySINK on the Unitree G1.

| Dataset | PHUMA Test | | | | | Unseen Video | | | | |
|---|---|---|---|---|---|---|---|---|---|---|
| | Total | Stationary | Angular | Vertical | Horizontal | Total | Stationary | Angular | Vertical | Horizontal |
| GMR | 84.0 | 92.1 | 77.8 | 77.1 | 89.1 | 75.2 | **99.1** | 77.8 | **61.7** | 52.7 |
| PhySINK | **89.9** | **94.2** | **87.6** | **84.2** | **91.8** | **81.7** | 97.4 | **86.7** | **61.7** | **71.4** |

In contrast, our proposed PhySINK method achieves both natural movement patterns and physical plausibility. The resulting motions maintain appropriate ankle angles while ensuring proper ground contact, demonstrating that PhySINK successfully balances motion naturalness with physical constraints. This improvement stems from the incorporation of explicit physical constraint terms in the optimization objective.

We also compare the motion retargeting results of GMR (Araujo et al., 2025), an optimization-based Inverse Kinematics approach, and PhySINK. The fundamental difference lies in how each method generates the intermediate target motion—transforming the source human motion to fit the humanoid's kinematic structure. GMR adapts the target motion using a heuristic scaling approach. It first estimates the source subject's height from the first component of the SMPL shape parameters ($\beta$). It then computes a scaling ratio between this estimated height and a pre-defined reference height (e.g., 1.8m). This ratio is applied as a multiplier to a set of manually tuned scaling factors for each limb. Mathematically, if the reference height is $H_{ref}$ and the subject's estimated height is $H_{src}$, the scaling ratio is $r = H_{src}/H_{ref}$. For a given limb (e.g., wrist) with a relative position vector $\mathbf{v}_{rel}$ from the pelvis and a manual scale factor $s_{limb}$, the scaled relative target becomes $\mathbf{v}'_{target} = s_{limb} \times r \times \mathbf{v}_{rel}$. The global target position is then reconstructed by adding this vector to the scaled global root position. In contrast, PhySINK adapts the target motion by optimizing the human shape itself to fit the humanoid. Instead of relying on scalar heuristics, PhySINK optimizes the SMPL shape parameters ($\beta$) to minimize the discrepancy between the human's keypoint positions and the humanoid's keypoints in a shared T-pose. These optimized shape parameters are then applied to the original motion sequence, preserving the original joint angles while naturally adjusting limb lengths to match the robot. Following target generation, the methods differ in execution: GMR utilizes a standard IK solver to track the scaled keypoints, whereas PhySINK finds optimal joint angles by minimizing the physics-informed losses described in the method section.

While GMR performs well for average-sized humans, its linear scaling heuristic fails to generalize across diverse human shapes. As shown in Figure 9, GMR struggles with significant height deviations. For example, for short humans, GMR generates undersized targets that force the robot into a crouched state (over-bent knees) even during standing motions. And for tall humans, GMR generates oversized targets beyond the robot's reach, causing the robot to walk with unnaturally locked knees and often leading to floating artifacts where feet lose contact with the ground. In contrast, PhySINK's approach—applying the original human motion's joint angles with an optimized, robot-matched shape—fully reflects the original motion's kinematics. This allows PhySINK to faithfully reproduce the intended motion and natural joint movements (e.g., natural knee flexion) regardless of the source subject's height or limb proportions, ensuring robust generalization.

To validate that GMR affects the training of motion tracking policy, we apply GMR to retarget the same motion sources used in PHUMA, excluding LaFAN1 (Harvey et al., 2020) and LocoMuJoCo (Al-Hafez et al., 2023), which are already pre-retargeted. We then train a MaskedMimic policy using the GMR-retargeted dataset, following the same training procedure described in the experiment section. As shown in Table 9, the policy trained with PhySINK-retargeted data achieves better imitation performance on both and Unseen Video benchmarks compared to the GMR-retargeted counterpart. This performance gap can be attributed to PhySINK's superior retargeting quality, as demonstrated in Table2.

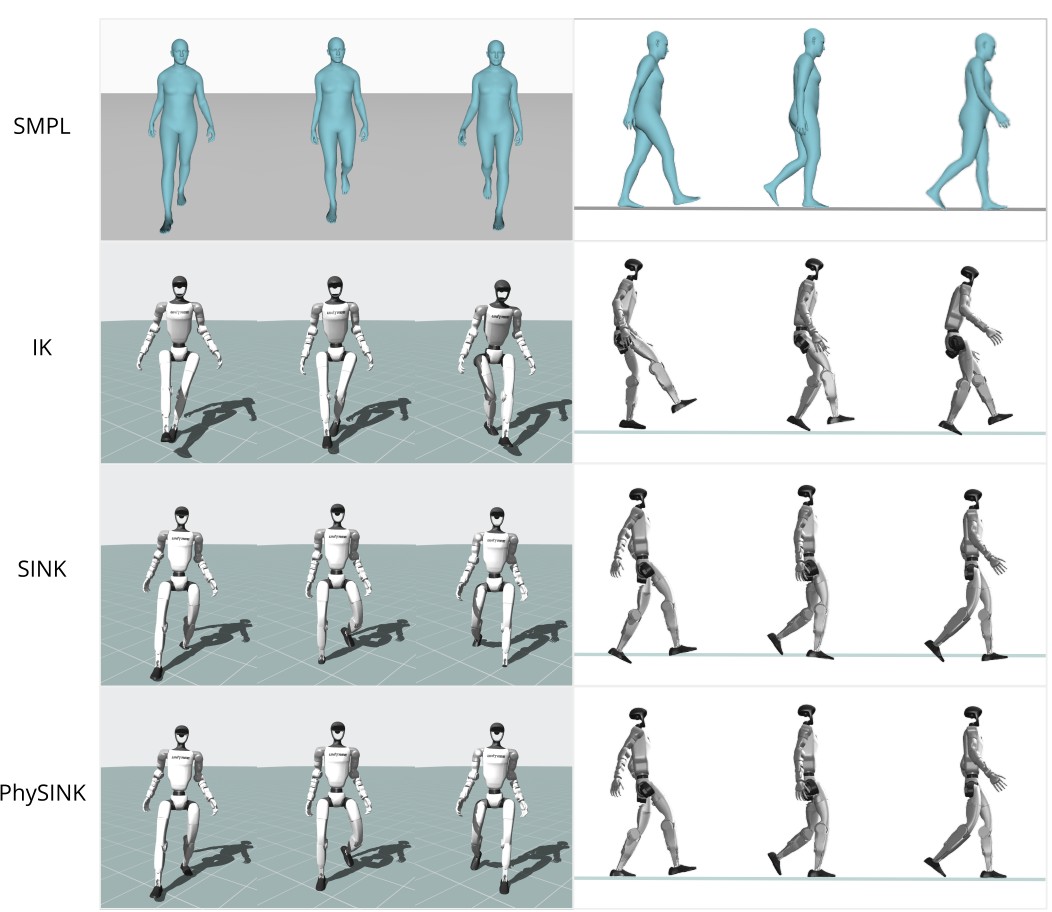

Figure 8: **Qualitative Comparison of Retargeting Methods**. This figure provides a visual comparison of human motion retargeted to a humanoid robot using the IK, SINK, and PhySINK methods. The top row shows the original human motion from the SMPL model, while the rows below show the resulting motions for each retargeting method.

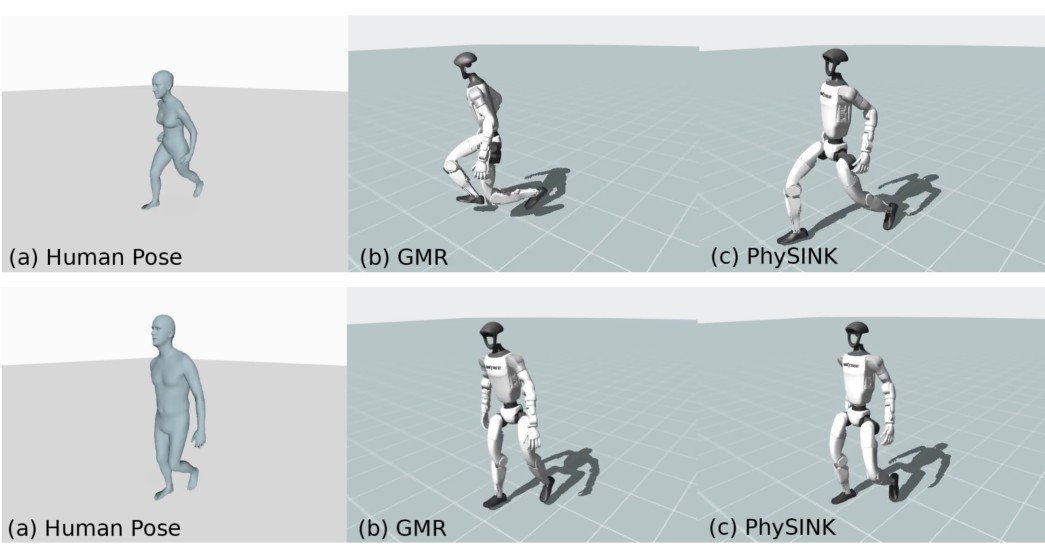

Figure 9: **Qualitative Comparison of Motion Retargeting: GMR vs. PhySINK.** Comparison showing the limitations of GMR when handling extreme human heights. The top row illustrates retargeting from a short subject, where GMR causes excessive motion compression and crouching. The bottom row illustrates retargeting from a tall subject, where GMR generates infeasible targets, resulting in artifacts like joint locking (e.g., rigid knees) and loss of ground contact. PhySINK maintains kinematic feasibility in both cases.

# B  DETAILS OF MOTION IMITATION LEARNING

## B.1  OBSERVATION SPACE COMPOSITIONS

This section provides detailed information about the observation space composition used in our experimental setup, as summarized in Table 10. The observation space consists of two main components: proprioceptive states and goal states.

**Proprioceptive States.** The proprioceptive information includes root height, body positions, body rotations, body velocities, and body angular velocities. The Unitree G1 and H1-2 robots have 33 and 25 bodies, respectively. For body positions, the root body is excluded from the position measurements.

**Goal States.** The goal states comprise both relative and absolute body positions and rotations. The relative component represents the difference between the future 15 timesteps of reference motion states and the current proprioceptive state. The absolute component represents states relative to the reference motion's root position, providing a root-relative coordinate frame for the target motion.

Table 10: Observation Space Dimensions

| State | Dimension G1 | Dimension H1-2 |
|---|---|---|
| **(a) Proprioceptive State** | | |
| Root height | 1 | 1 |
| Body position | $32 \times 3$ | $24 \times 3$ |
| Body rotation | $33 \times 6$ | $25 \times 6$ |
| Body velocity | $33 \times 3$ | $25 \times 3$ |
| Body angular velocity | $33 \times 3$ | $25 \times 3$ |
| **(b) Goal State** | | |
| Relative body position | $33 \times 15 \times 3$ | $25 \times 15 \times 3$ |
| Absolute body position | $33 \times 15 \times 3$ | $25 \times 15 \times 3$ |
| Relative body rotation | $33 \times 15 \times 6$ | $25 \times 15 \times 6$ |
| Absolute body rotation | $33 \times 15 \times 6$ | $25 \times 15 \times 6$ |
| Time | $33 \times 15 \times 1$ | $25 \times 15 \times 1$ |
| **Total dim** | 9898 | 7498 |

## B.2  REWARD FUNCTION

The reward function used for training the tracking policy consists of multiple components, as detailed in Table 11. The overall reward structure comprises two main categories: motion tracking task rewards and regularization rewards.

**Motion Tracking Rewards.** These components encourage the policy to match the reference motion by providing higher rewards when the robot's proprioceptive states closely resemble the target motion states.

**Regularization Rewards.** To promote smooth and stable motion execution, we include regularization terms that penalize undesirable behaviors. Specifically, we augment the standard MaskedMimic reward formulation with action rate penalties that discourage large changes between consecutive actions, helping to ensure smooth joint movements and prevent abrupt motion transitions.

## B.3  PPO HYPERPARAMETER

The detailed hyperparameter configuration used for PPO training is provided in Table 12.

Table 11: Reward function terms for training

| Term | Expression | Weight |
|------|-----------|--------|
| **(a) Task** | | |
| Global body position | $\exp(-100 \cdot \|p_t - \hat{p}_t\|_2^2)$ | 0.5 |
| Root height | $\exp(-100 \cdot (h_t^{\text{root}} - \hat{h}_t^{\text{root}})^2)$ | 0.2 |
| Global body rotation | $\exp(-10 \cdot \|\theta_t \ominus \hat{\theta}_t\|_2^2)$ | 0.3 |
| Global body velocity | $\exp(-0.5 \cdot \|v_t - \hat{v}_t\|_2^2)$ | 0.1 |
| Global body angular velocity | $\exp(-0.1 \cdot \|\omega_t - \hat{\omega}_t\|_2^2)$ | 0.1 |
| **(b) Regularization** | | |
| Power consumption | $\|F \odot \dot{q}\|_1$ | -1e-05 |
| Action rate | $\|a_t - a_{t-1}\|_2^2$ | -0.2 |

Table 12: PPO Hyperparameter Values for Model Training

| Hyperparameter | Value |
|----------------|-------|
| Optimizer | Adam |
| Num envs | 8192 |
| Mini Batches | 32 |
| Learning epochs | 1 |
| Entropy coefficient | 0.0 |
| Value loss coefficient | 0.5 |
| Clip param | 0.2 |
| Max grad norm | 50.0 |
| Init noise std | -2.9 |
| Actor learning rate | 2e-5 |
| Critic learning rate | 1e-4 |
| GAE decay factor($\lambda$) | 0.95 |
| GAE discount factor($\gamma$) | 0.99 |
| Actor Transformer dimension | 512 |
| Actor layers | 4 |
| Actor heads | 4 |
| Critic MLP size | [1024, 1024, 1024, 1024] |
| Activation | ReLU |

# C EXPERIMENT DETAILS

## C.1 SELF-COLLECTED VIDEO DATASET

To ensure fair evaluation of imitation performance on unseen motions, we create a custom evaluation dataset using self-collected video recordings. This dataset contains motions uniformly distributed across the 11 motion types shown in Figure 7b, providing balanced coverage for comprehensive performance assessment.

The dataset creation process follows three main steps: (1) recording videos of human performers executing each motion type, (2) converting videos into SMPL human motion parameters using a video-to-motion model, and (3) retargeting the human motions to humanoid robot motions using our PhySINK method.

First, we record videos covering all 11 motion categories, collecting a uniform distribution for each type. We then apply the TRAM video-to-motion model (Wang et al., 2024) to extract SMPL motion parameters from the recorded videos. Finally, we process these SMPL motions with PhySINK retargeting to generate physically plausible humanoid motions. Example results from this dataset are illustrated in Figure 10.

This self-collected evaluation set ensures that our performance assessments are conducted on completely unseen motions that were not influenced by any training data sources, providing an unbiased evaluation of generalization capabilities.

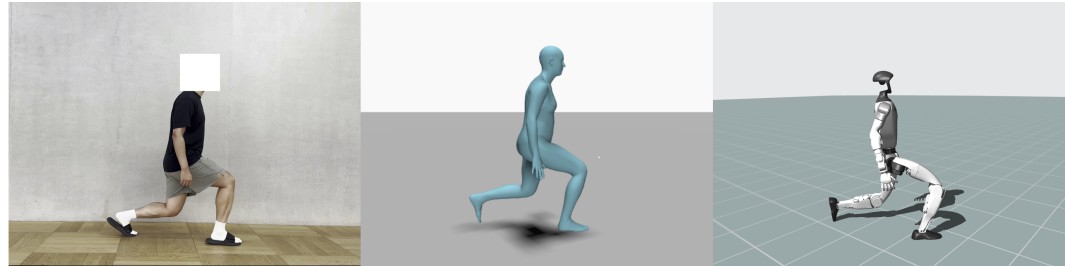

Figure 10: **Overview of the Self-collected Data Pipeline**. This figure illustrates the three main steps of our data collection pipeline: (left) a self-recorded video of a human motion, (center) the motion extracted using a video-to-motion model, and (right) the final motion retargeted to a humanoid robot.

## C.2 SUCCESS RATE THRESHOLD ANALYSIS

To demonstrate the limitations of the conventional success rate threshold, we evaluate imitation performance using both the standard 0.5m threshold and our proposed stricter 0.15m threshold. This comparison reveals the true quality differences between policies trained on different datasets.

Tables 13 and 14 present the results for both threshold settings. Under the loose 0.5m threshold, policies trained on different datasets show relatively similar success rates, with differences appearing modest. However, when evaluated with the stricter 0.15m threshold, performance differences become substantially more pronounced.

These results confirm that PHUMA-trained policies achieve more precise motion tracking, producing imitations that remain accurate even under stringent evaluation criteria. The threshold analysis validates our choice to adopt the 0.15m threshold as a more meaningful measure of imitation quality.

## C.3 CROSS-SIMULATOR GENERALIZATION ANALYSIS

To evaluate whether cross-simulator transfer, we trained the KungfuBot(Han et al., 2025) policy in Isaac Gym using an MLP architecture, which has demonstrated strong sim-to-real transfer capabilities. We compared two training configurations: one using AMASS with SINK retargeting and another using PHUMA. For both configurations, we followed the standard KungfuBot (Han et al.,

Table 13: Performance Comparison based on Success Threshold in PHUMA Test

| Dataset | Hours | Success Threshold=0.15m | | | | | Success Threshold=0.5m | | | | |
|---|---|---|---|---|---|---|---|---|---|---|---|
| | | Total | Stationary | Angular | Vertical | Horizontal | Total | Stationary | Angular | Vertical | Horizontal |
| **(a) G1** | | | | | | | | | | | |
| LaFAN1 | 2.4 | 46.1 | 66.1 | 36.2 | 24.0 | 42.5 | 74.8 | 87.8 | 69.2 | 47.1 | 72.6 |
| AMASS | 20.9 | 76.2 | 88.5 | 72.1 | 56.8 | 66.8 | 90.2 | 95.0 | 87.9 | 81.1 | 83.7 |
| Humanoid-X | 231.4 | 50.6 | 78.4 | 43.0 | 26.0 | 31.8 | 78.4 | 91.3 | 72.9 | 59.5 | 65.9 |
| **PHUMA** | 73.0 | **92.7** | **95.6** | **91.7** | **86.0** | **85.6** | **97.1** | **98.7** | **96.5** | **94.4** | **92.5** |
| **(b) H1-2** | | | | | | | | | | | |
| LaFAN1 | 2.4 | 62.0 | 79.3 | 54.7 | 26.6 | 58.9 | 70.8 | 92.4 | 66.7 | 56.4 | 68.2 |
| AMASS | 20.9 | 54.4 | 74.9 | 45.9 | 17.2 | 49.6 | 70.4 | 86.3 | 62.6 | 41.4 | 65.9 |
| Humanoid-X | 231.4 | 49.7 | 74.6 | 40.4 | 17.0 | 37.3 | 54.8 | 78.5 | 45.2 | 22.1 | 43.2 |
| **PHUMA** | 73.0 | **82.7** | **91.5** | **79.5** | **68.1** | **68.4** | **92.0** | **96.6** | **89.7** | **85.6** | **79.4** |

Table 14: Performance Comparison based on Success Threshold in Unseen Video

| Dataset | Hours | Success Threshold=0.15m | | | | | Success Threshold=0.5m | | | | |
|---|---|---|---|---|---|---|---|---|---|---|---|
| | | Total | Stationary | Angular | Vertical | Horizontal | Total | Stationary | Angular | Vertical | Horizontal |
| **(a) G1** | | | | | | | | | | | |
| LaFAN1 | 2.4 | 28.4 | 46.9 | 28.4 | 19.6 | 10.5 | 78.2 | 85.5 | 70.8 | 76.3 | 80.8 |
| AMASS | 20.9 | 70.2 | 90.7 | 75.0 | 62.7 | 44.1 | 92.3 | 99.2 | 92.1 | 82.1 | 88.0 |
| Humanoid-X | 231.4 | 39.1 | 78.0 | 39.6 | 23.0 | 6.5 | 84.1 | 98.3 | 79.9 | 76.0 | 76.2 |
| **PHUMA** | 73.0 | **82.9** | **96.7** | **88.0** | **71.8** | **67.1** | **93.7** | **100.0** | **96.8** | **85.9** | **84.7** |
| **(b) H1-2** | | | | | | | | | | | |
| LaFAN1 | 2.4 | 70.8 | 92.4 | 66.7 | 56.4 | 68.2 | 85.5 | 97.5 | 79.0 | 77.5 | 90.0 |
| AMASS | 20.9 | 64.3 | 87.3 | 59.7 | 46.0 | 63.9 | 80.4 | 93.3 | 69.9 | 72.8 | 89.0 |
| Humanoid-X | 231.4 | 60.5 | 88.3 | 60.0 | 48.7 | 39.7 | 68.7 | 93.3 | 65.1 | 60.2 | 50.5 |
| **PHUMA** | 73.0 | **78.6** | **97.5** | **76.8** | **74.5** | **63.8** | **89.9** | **99.2** | **89.4** | **84.6** | **83.9** |

2025) teacher-student training pipeline and evaluation metrics. Tables 15 and 16 present the teacher and student policy performance in the training simulator (Isaac Gym). The results show that policies trained with PHUMAachieve superior imitation performance, demonstrating that PHUMA's benefits extend to other motion imitation algorithms. Importantly, this performance advantage also transfers to the evaluation simulator. Table 6 shows the zero-shot performance in MuJoCo, where the PHUMA-trained policy maintains its superiority across motion categories, indicating robust cross-simulator generalization.

# D ABLATION STUDIES OF DATASET

## D.1 MOCAP ONLY AND VIDEO ONLY DATA PERFORMANCE

To analyze the influence of the human motion source on the downstream motion tracking policy, we divide the PHUMA dataset into two distinct subsets: motions derived from motion capture (Mocap) and motions derived from video-to-motion estimation (Video-Sourced).

We leverage these subsets to train the MaskedMimic policy using identical hyperparameters in Section B. As demonstrated in Table 18, the policy trained with video-sourced PHUMA consistently yielded superior imitation performance across all motion categories compared to the policy trained with the mocap-sourced subset.

We attribute this result primarily to the significantly larger and broader motion distribution of the video-sourced data. As illustrated in Figure 7a, the video-sourced dataset covers a much broader range of motion types and contains nearly two times more data than the mocap-sourced dataset. Furthermore, as shown in Table 17, the PhySINK retargeting method ensures competitive motion quality for both subsets. Because the retargeting quality is similar, the dominant factor leading to the higher imitation performance is the larger size and greater diversity of the video-sourced dataset.

## D.2 PHYSICS-BASED FILTERING

This section provides ablation studies on the physics-based filtering criteria used in data curation (Table 8) and the physics-constrained losses used in PhySINK.

Table 15: **Imitation Performance of the Kungfubot Teacher Policy in Isaac Gym.** We evaluate the imitation performance of Kungfubot teacher policy with unseen motions on the Unitree G1.

| Dataset | PHUMA Test | | | | | Unseen Video | | | | |
|---|---|---|---|---|---|---|---|---|---|---|
| | Total | Stationary | Angular | Vertical | Horizontal | Total | Stationary | Angular | Vertical | Horizontal |
| AMASS | 77.8 | 90.8 | 71.2 | 58.3 | 84.7 | 83.5 | **100.0** | 83.3 | 74.5 | 72.5 |
| PHUMA | **91.0** | **96.4** | **88.3** | **87.6** | **90.5** | **87.1** | 99.1 | **89.2** | **77.7** | **76.9** |

Table 16: **Imitation Performance of the Kungfubot Student Policy in Isaac Gym.** We evaluate the imitation performance of Kungfubot teacher policy with unseen motions on the Unitree G1.

| Dataset | PHUMA Test | | | | | Unseen Video | | | | |
|---|---|---|---|---|---|---|---|---|---|---|
| | Total | Stationary | Angular | Vertical | Horizontal | Total | Stationary | Angular | Vertical | Horizontal |
| AMASS | 66.6 | 86.7 | 58.8 | 41.8 | 68.4 | 67.7 | **97.4** | 68.0 | 59.6 | 37.4 |
| PHUMA | **82.9** | **93.5** | **78.5** | **74.2** | **81.7** | **73.8** | **97.4** | **76.9** | **63.8** | **47.3** |

Table 17: **Performance Evaluation of PHUMA based on Data Source (MoCap vs. Video).** We present a quantitative comparison evaluating the performance achieved using PHUMA data derived from motion capture (MoCap) versus video, concluding that both sources offer competitive results.

| | Motion Fidelity (%) | Joint Feasibility (%) | Non-Floating (%) | Non-Penetration (%) | Non-Skating (%) |
|---|---|---|---|---|---|
| MoCap | **96.7** | **100.0** | **99.9** | 94.3 | **92.1** |
| Video | 93.8 | **100.0** | **99.9** | **98.0** | 88.3 |

### D.2.1 DATA DISTRIBUTION BASED ON PHYSICS-BASED FILTERING

Figure 11 shows the filtering statistics when sequentially applying the physics-based criteria from Table 8 to Humanoid-X (Mao et al., 2025). The filters are applied in the following order: (1) root jerk filter (jerk $< 50\text{m/s}^3$), (2) contact filter (foot contact score $> 0.6$), (3) height filter (minimum pelvis height $> 0.6$m and maximum pelvis height $< 1.5$m), and (4) base of support (BoS) filter (pelvis distance to BoS $< 6$cm and spine1 distance to BoS $< 11$cm). After applying all filters sequentially, 27.1% of the original Humanoid-X dataset remains, representing motions that satisfy physical plausibility constraints.

### D.2.2 PHYSINK'S ROBUSTNESS TO NOISY MOTION SOURCES

To evaluate how robustly PhySINK handles noisy human motion inputs, we retarget Humanoid-X motion sources with varying levels of filtering: (1) raw Humanoid-X (no filtering), (2) Humanoid-X + jerk filtering, (3) Humanoid-X + foot contact filtering, (4) Humanoid-X + height filtering, (5) Humanoid-X + BoS filtering, and (6) Humanoid-X + all filters. We then apply PhySINK to each variant. Note that we exclude pre-retargeted datasets (LaFAN1 and LocoMuJoCo) from this analysis to isolate the effect of filtering on Humanoid-X. As shown in Table 19, PhySINK demonstrates robust retargeting performance across motion sources with varying noise levels, successfully handling physical implausibilities present in the raw data.

### D.2.3 EFFECT OF PHYSICS-BASED FILTERING ON IMITATION PERFORMANCE

To evaluate how physics-based filtering affects motion tracking performance, we train MaskedMimic policies using the datasets described in Section D.2.2, following the same training protocol described in the experiment section. As shown in Table 20, policies trained on datasets with at least one physics-based filter achieve better imitation performance compared to those trained on unfiltered data. Furthermore, applying all filters to Humanoid-X yields the best performance, demonstrating the importance of physics-based motion curation for learning high-quality imitation policies.

### D.3 IMPACT OF MOTION RETARGETING QUALITY ON POLICY PERFORMANCE

To investigate how physical artifacts in retargeted motion data affect policy learning, we train MaskedMimic policies on datasets generated using six retargeting methods with varying artifact

Table 18: **Imitation Performance of PHUMA based on Data Source (MoCap vs. Video).** We evaluate the imitation performance of MaskedMimic policy trained with PHUMA data derived from motion capture (MoCap) versus video on the Unitree G1.

| Dataset | PHUMA Test | | | | | Unseen Video | | | | |
|---------|-------|------------|---------|----------|------------|-------|------------|---------|----------|------------|
| | Total | Stationary | Angular | Vertical | Horizontal | Total | Stationary | Angular | Vertical | Horizontal |
| MoCap | 75.2 | 88.2 | 68.4 | 49.0 | 86.9 | 73.0 | 96.6 | 73.9 | 56.4 | 58.2 |
| Video | **85.7** | **92.9** | **81.8** | **75.5** | **89.5** | **76.2** | **100.0** | **76.4** | **59.6** | **62.6** |

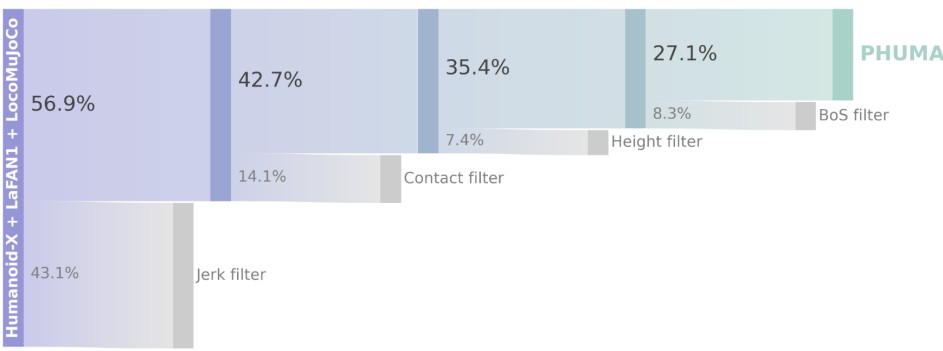

Figure 11: **Dataset Statistics After Physics-based Filtering.** Distribution of motion sequences after applying physics-based filtering to the combined Humanoid-X, LaFAN1, and LocoMuJoCo datasets.

Table 19: **PhySINK Retargeting Robustness to Noisy Motion Sources.** This figure presents an experiment to evaluate how robustly the PhySINK method retargets various noisy Humanoid-X motion sources. The six distinct motion source groups used for retargeting are compared: (1) Original Humanoid-X, and Humanoid-X motion sources sequentially refined by applying (2) Jerk filtering, (3) Foot Contact filtering, (4) Height filtering, (5) BoS filtering, and (6) All filtering (PHUMA).

| Motion Source | Hours | Motion Fidelity (%) | Joint Feasibility (%) | Non-Floating (%) | Non-Penetration (%) | Non-Skating (%) |
|---------------|-------|---------------------|-----------------------|------------------|---------------------|-----------------|
| Humanoid-X | 237.2 | 70.6 | **100.0** | 98.7 | 92.2 | 90.1 |
| Jerk Filter | 141.1 | 76.9 | **100.0** | 99.7 | 96.5 | **92.0** |
| Foot Contact Filter | 123.1 | 77.1 | **100.0** | 99.8 | **96.8** | 90.6 |
| Height Filter | 135.7 | 87.0 | **100.0** | 99.4 | 94.7 | 89.8 |
| BoS Filter | 110.3 | 90.7 | **100.0** | 99.4 | 94.9 | 90.0 |
| All Filter | 62.2 | **94.8** | **100.0** | **99.9** | 96.7 | 89.7 |

levels (Table 2). The methods are: (1) **IK**, which produces significant artifacts in motion fidelity, joint limits, grounding, and skating; (2) **GMR**, which reduces motion fidelity loss, grounding issues, and skating compared to IK; (3) **SINK**, which improves motion fidelity and joint limit violations; (4) **SINK + Joint Feasibility Loss**, which further reduces joint limit violations; (5) **SINK + Joint Feasibility + Grounding Loss**, which addresses all artifact types except skating; and (6) **PhySINK**, which minimizes all physical artifacts. To isolate the effect of retargeting quality, we exclude the LaFAN1 and LocoMuJoCo datasets from this analysis, as they were pre-retargeted and would not allow for fair comparison across methods.

In Table 21, our results show that SINK-based methods, which first optimize the humanoid body shape before applying it to the original motion, consistently outperform IK-based methods that rely on heuristic scaling to bridge human-humanoid discrepancies. Notably, GMR achieves better performance than IK despite having similar joint limit issues, while the SINK variants further outperform GMR in tracking performance. However, across the SINK variants themselves (SINK, SINK + Joint Feasibility Loss, SINK + Joint Feasibility + Grounding Loss), we observe comparable performance despite their similar levels of motion fidelity, joint feasibility, and grounding quality—the main differences among these variants being penetration and skating artifacts.

These results indicate that motion fidelity is the most critical factor affecting motion tracking performance. The substantial performance improvement of GMR over IK—achieved primarily through motion fidelity gains rather than joint feasibility improvements—demonstrates that preserving mo-

Table 20: **Imitation Performance on Diverse Motion Sources.** We evaluate the imitation performance of MaskedMimic policies trained on datasets with varying noise levels (described in Section D.2.2) using the Unitree G1 robot.

| Motion Source | PHUMA Test | | | | | Unseen Video | | | | |
|---|---|---|---|---|---|---|---|---|---|---|
| | Total | Stationary | Angular | Vertical | Horizontal | Total | Stationary | Angular | Vertical | Horizontal |
| Humanoid-X | 85.7 | 92.6 | 82.2 | 75.5 | 89.5 | 75.4 | 96.6 | 81.3 | 58.5 | 52.7 |
| Jerk Filter | 87.1 | 93.1 | 83.2 | 80.4 | 90.9 | 78.8 | **100.0** | 76.8 | **64.9** | 70.3 |
| Foot Contact Filter | 88.2 | 93.9 | 84.9 | 81.8 | 90.5 | 78.6 | 95.7 | 82.3 | 61.7 | 65.9 |
| Height Filter | 86.9 | 93.0 | 83.2 | 81.6 | 88.9 | 77.2 | 94.8 | 79.8 | 59.6 | 67.0 |
| BoS Filter | 86.8 | 92.7 | 83.6 | 79.1 | 89.8 | 77.0 | 96.6 | 79.8 | 61.7 | 61.5 |
| ALL Filter | **89.9** | **94.2** | **87.6** | **84.2** | **91.8** | **81.7** | 97.4 | **86.7** | 61.7 | **71.4** |

Table 21: **Ablation Studies of Imitation Performance on Retargeting Loss.** We evaluate the imitation performance of MaskedMimic policies trained with and without the physical constraint loss (Table 2) using the Unitree G1 robot.

| Dataset | PHUMA Test | | | | | Unseen Video | | | | |
|---|---|---|---|---|---|---|---|---|---|---|
| | Total | Stationary | Angular | Vertical | Horizontal | Total | Stationary | Angular | Vertical | Horizontal |
| IK | 70.5 | 85.9 | 63.3 | 47.6 | 77.3 | 68.5 | 96.6 | 72.4 | 43.6 | 49.5 |
| GMR | 84.0 | 92.1 | 77.8 | 77.1 | 89.1 | 75.2 | 99.1 | 77.8 | 61.7 | 52.7 |
| SINK | 89.1 | 94.0 | 86.0 | 84.9 | 90.7 | 79.0 | **100.0** | 81.8 | **62.8** | 62.6 |
| + Joint Feasibility Loss | 87.0 | 92.1 | 83.6 | 79.8 | 90.9 | 78.6 | 94.8 | 84.7 | 56.4 | 67.0 |
| + Grounding Loss | **90.0** | 93.5 | **87.9** | **85.6** | **92.0** | 80.4 | 98.3 | 86.2 | 60.6 | 64.8 |
| + Skating Loss | 89.9 | **94.2** | 87.6 | 84.2 | 91.8 | **81.7** | 97.4 | **86.7** | 61.7 | **71.4** |

tion fidelity during retargeting has a greater impact on policy learning than other artifact types. However, once motion fidelity reaches a certain threshold (as in the SINK variants), further reductions in other artifact types yield diminishing returns for policy learning.

