# OpenReview forum: "PHUMA: Physically-Grounded Humanoid Locomotion Dataset"
_ICLR.cc/2026/Conference — Submitted to ICLR 2026_

### Official Review · Reviewer_nr5V · 2025-10-26

**Soundness:** 3
**Presentation:** 3
**Contribution:** 3
**Rating:** 6
**Confidence:** 3

**Summary:**

This submission introduces PHUMA, a physically-grounded humanoid locomotion dataset created from human videos with a physics-constrained retargeting pipeline that enforces joint limits, reliable ground contact, and removes foot-skate. Trained policies on PHUMA outperform counterparts trained on AMASS and Humanoid-X in motion imitation and path-following tests. The authors plan to release the dataset to the community.

**Strengths:**

- A large-scale (73h), curated, physics-aware dataset that  addresses well-known artifacts in video-derived motion (e.g., joint limits, penetration, floating, foot-skate). This provides practical value to the community.
- The method is sound. The proposed PhySINK extends SINK to deal with joint limits, ground contact, and foot skating.
- The evaluation is principled and thorough. Evaluations show consistent gains over AMASS and Humanoid-X across two use cases (unseen-video imitation, pelvis-only path following).

**Weaknesses:**

- Related works: Please compare/relate to recent efforts that also scale human to humanoid data, e.g., H2O/Human2Humanoid, ASAP, Humanoid Policy ~ Human Policy.
- How does the proposed method handle non-planar ground, such as ground with height changes or stairs?
- Physically invalid motions are removed but not refined. This could reduce the diversity of the data.

**Questions:**

- what is the data distribution in terms of scene geometry, material, and semantics diversity (e.g., indoor vs outdoor, height variation, ground material)
- To what extent is retargeting tied to a specific humanoid morphology? What is the distribution over body shape and heights.
- How is foot contact detected? How reliable is that?
- It would strengthen the paper to quantify how each physics constraint (contact, joint limits, anti-skate) contributes to downstream policy quality.

---

> ### Author Response · Authors · 2025-11-27
>
> Hi Reviewer nr5V
> Thank you for the contructive feedback.
>
> > W1: Related works: Please compare/relate to recent efforts that also scale human to humanoid data, e.g., H2O/Human2Humanoid, ASAP, Humanoid Policy ~ Human Policy.
>
> We thank the reviewer for highlighting these important recent works. We have revised the Related Work to explicitly compare/relate H2O, ASAP, and Humanoid Policy ~ Human Policy with our approach.
>
> Specifically, we acknowledged H2O for introducing shape-adaptive approach to address morphological mismatches. We also discussed the "sim-to-data" cleaning strategy of ASAP, contrasting its reliance on tracking policies with our focus on physical validity, while noting its application to agile locomotion. Finally, we referenced Humanoid Policy ~ Human Policy as a key example of leveraging human video for humanoid manipulation tasks.
>
> > W2: How does the proposed method handle non-planar ground, such as ground with height changes or stairs?
>
> In this work we only consider to make the physically reliable retaregt dataset on the planar ground. Making physically reliable humanoid on non-plannar ground like stair or with objects might be very interesting future work.
>
> > W3: Physically invalid motions are removed but not refined. This could reduce the diversity of the data.
>
> We agree this is an important consideration. As demonstrated in our ablation studies (see general response #3), data quality is more critical than data quantity for learning robust motion tracking policies. While our current approach focuses on curating existing datasets through physics-based filtering, extending these curation principles to refine general motion datasets into physically reliable training data represents a promising direction for future work.
>
> > Q1: what is the data distribution in terms of scene geometry, material, and semantics diversity (e.g., indoor vs outdoor, height variation, ground material)
>
> We have updated Table 1 to include information on indoor vs. outdoor scenes. Since we focus on whole-body motions on flat terrain, our dataset does not contain paired scenes with varying ground heights or materials. We believe constructing motion datasets paired with scenes featuring non-planar ground and diverse materials is an interesting future direction.
>
> > Q2: To what extent is retargeting tied to a specific humanoid morphology? What is the distribution over body shape and heights.
>
> PhySINK operates in two stages: (1) generating target motions adapted to the humanoid's body shape, and (2) optimizing humanoid joint angles to match the target motions using multiple physics-based constraint losses.
>
> In the first stage, we optimize the SMPL shape parameters ($\beta$) to find the body shape that best matches the target humanoid in a T-pose. We then substitute these optimized $\beta$ parameters into the original motion sequence while preserving the original joint angles and kinematic structure. This ensures that the target motion reflects the humanoid's body proportions while maintaining the movement characteristics of the source motion. Importantly, for a given humanoid robot, the optimized $\beta$ parameters remain fixed and can be reused across all motions, making the retargeting process efficient for large-scale datasets.
>
> > Q3: How is foot contact detected? How reliable is that?
>
> We use a data-driven method based on majority voting with the foot mesh vertices. We determine the ground level by finding where the foot vertices vote for the ground surface. We vertically shift the human mesh so that the voted ground height becomes 0. Then, we detect contact by counting the number of foot vertices within a $\pm 2.5$ cm threshold band.
>
> We believe this is more reliable than existing approaches that often set a fixed threshold for ankle height to estimate contact masks. Those approaches are susceptible to noise, especially for motions reconstructed from videos rather than motion capture, where foot floating or penetration are common issues. On the other hand, since we align the foot mesh with the ground first using a data-driven approach and then estimate the contact mask, our method handles these artifacts robustly.
>
> > Q4: It would strengthen the paper to quantify how each physics constraint (contact, joint limits, anti-skate) contributes to downstream policy quality.
>
> We agree this is an important analysis. To address your comment, we have conducted experiments to investigate how retargeting quality affects policy performance. Please refer to the general response #4 for detailed explanations and results.

---

### Official Review · Reviewer_QDRY · 2025-10-30

**Soundness:** 2
**Presentation:** 2
**Contribution:** 3
**Rating:** 4
**Confidence:** 4

**Summary:**

To address the data scalability issue of humanoid locomotion, the authors proposed PHUMA. With carefully curated physics-aware motion filtering, retargetting, and learning, PHUMA managed to effectively enlarge the available data scale for humanoid. The effectiveness of PHUMA is validated on unseen motion imitation and path following. Impressive performance is achieved.

**Strengths:**

- The curated PHUMA is superior to existing efforts in scale and diversity.

- The proposed pipeline seems reasonable and practical.

- The proposed pipeline and dataset could serve as valuable resources for humanoid learning.

**Weaknesses:**

- A noticeable characteristic of PHUMA is its heterogeneous data sources. Extending the existing data quality analysis and performance analysis to data from different sources would be preferred. A separate evaluation of mocap-sourced and video-sourced PHUMA would be helpful.

- Comparisons between the proposed physink and related works, like GMR, are missing.

- Though the comparison of heuristic metrics in Table 2 is straightforward, a missed chance would be investigating the relationship between these physical artifacts and their influence on the humanoid learning procedure.

- The physics-based filtering hasn't been well analyzed with ablation studies.

- Many details are missing, some of which are closely related to the data quality and algorithm performance.

- No video demonstration is attached, which is important for identifying the data quality.

- No hardware deployment results are provided.

**Questions:**

- In Table 1, most of the video datasets only have a frame rate of 30FPS. How would the low-pass filter with a cutoff frequency of 30Hz, as mentioned in A.1.1, work for these data?

- How were the limits and thresholds in Table 7 decided? How many motion sequences were filtered out for each term? Also, how were the corresponding influences on the performance?

- If the curation pipeline is applied to LAFAN1 and AMASS, would the corresponding performance be improved?

---

> ### Author Response · Authors · 2025-11-27
>
> Hello Reviewer QDRY,
>
> Thank you for your thorough review and excellent questions. We appreciate you highlighting areas for deeper analysis and clarification. We have prepared detailed answers addressing all your points, including the requested ablation studies and analyses.
>
> > W1: A noticeable characteristic of PHUMA is its heterogeneous data sources. Extending the existing data quality analysis and performance analysis to data from different sources would be preferred. A separate evaluation of mocap-sourced and video-sourced PHUMA would be helpful.
>
> We agree this is a valuable analysis. To investigate the impact of data source on policy performance, we trained separate MaskedMimic policies using the mocap-only subset and the video-sourced subset of PHUMA.
>
> As shown in Table 18,
>
> | Dataset | **PHUMA Test** | | | | | **Unseen Video** | | | | |
> | :--- | :---: | :---: | :---: | :---: | :---: | :---: | :---: | :---: | :---: | :---: |
> | | Total | Stationary | Angular | Vertical | Horizontal | Total | Stationary | Angular | Vertical | Horizontal |
> | MoCap | 75.2 | 88.2 | 68.4 | 49.0 | 86.9 | 73.0 | 96.6 | 73.9 | 56.4 | 58.2 |
> | Video | **85.7** | **92.9** | **81.8** | **75.5** | **89.5** | **76.2** | **100.0** | **76.4** | **59.6** | **62.6** |
>
> the policy trained on the video-sourced dataset achieves better imitation performance compared to the mocap-only dataset across all motion categories. We attribute this improvement to two factors:
> - The video-sourced dataset is approximately twice the size of the mocap-sourced dataset and covers a broader motion distribution (Figure 7a).
> - Both datasets exhibit similar retargeting quality (Table 17).
>
> | | Motion Fidelity (%) | Joint Feasibility (%) | Non-Floating (%) | Non-Penetration (%) | Non-Skating (%) |
> | :--- | :---: | :---: | :---: | :---: | :---: |
> | MoCap | **96.7** | **100.0** | **99.9** | 94.3 | **92.1** |
> | Video | 93.8 | **100.0** | **99.9** | **98.0** | 88.3 |
>
> Given the comparable retargeting quality, the superior performance of the video-sourced policy can be attributed to its larger dataset size and broader motion coverage. This demonstrates that when retargeting quality is held constant, increasing dataset scale and diversity leads to improved motion tracking performance. Please refer to Appendix D.1 for detailed explanations.
>
> > W2: Comparisons between the proposed physink and related works, like GMR, are missing.
>
> We fully agree that comparison with GMR is essential. We have performed a comprehensive comparison, including both quantitative and qualitative results, detailed in our general response #2. Please refer the general response #2 for detailed explanations.
>
> > W3: Though the comparison of heuristic metrics in Table 2 is straightforward, a missed chance would be investigating the relationship between these physical artifacts and their influence on the humanoid learning procedure.
>
> We agree this is an important analysis. To address your comment, we have conducted a thorough investigation of how motion retargeting quality impacts policy performance. We provide detailed explanations and results in the general response #4. Please refer to the general response #4 for complete details.
>
> > W4: The physics-based filtering hasn't been well analyzed with ablation studies. Many details are missing, some of which are closely related to the data quality and algorithm performance.
> >  W5: Many details are missing, some of which are closely related to the data quality and algorithm performance.
>
> We agree this is an important analysis. In response, we have conducted a thorough investigation of how each filtering component affects both the retargeting quality of PhySINK and the downstream policy performance. Please refer to the general response #3 for detailed explanations.
>
>
> > W6: No video demonstration is attached, which is important for identifying the data quality.
>
> We fully agree with your suggestion. We have added comprehensive video demonstrations to our supplementary page: link  ([anonymous.video_link](https://anonymous-robotics-researcher.github.io/Paper_10482/)). These include examples of the PHUMA dataset motions, qualitative comparisons of retargeting methods, and motion tracking policy performance.
>
> > W7: No hardware deployment results are provided.
>
> As detailed in the general response #5, we acknowledge that sim-to-real deployment is the most important validation. Due to current hardware availability constraints, we have provided an alternative validation: extensive quantitative and qualitative sim-to-sim performance using the established KungfuBot2 policy across Isaac Gym and MuJoCo.

---

> ### Author Response · Authors · 2025-11-27
>
> > Q1: In Table 1, most of the video datasets only have a frame rate of 30FPS. How would the low-pass filter with a cutoff frequency of 30Hz, as mentioned in A.1.1, work for these data?
>
> We thank the reviewer for identifying this confusion. We apologize for the misleading notation in the original draft. The "$f_s = 30\text{ Hz}$" in Appendix A.1.1 referred to the sampling frequency of the motion data, rather than the cutoff frequency of the filter. As the reviewer correctly noted, a cutoff of 30 Hz would be invalid for 30 FPS data. The actual cutoff frequencies used were 3 Hz for root translation and 6 Hz for body pose/orientation. We have revised Appendix A.1.1 to clarify this point.
>
> > Q2: How were the limits and thresholds in Table 7 decided? How many motion sequences were filtered out for each term? Also, how were the corresponding influences on the performance?
>
> Figure 11 shows the percentage of motions retained after applying each physics-based filtering component (filtering criteria are detailed in Table 8). As discussed in the general response #3, policies trained with at least one filtering component consistently outperform those trained on unfiltered data, and applying all filters yields the best imitation performance. This demonstrates that data quality improvements from physics-based filtering outweigh the reduction in dataset size.
>
> > Q3: If the curation pipeline is applied to LAFAN1 and AMASS, would the corresponding performance be improved?
>
> The performance results for LAFAN1 and AMASS reported in the paper are based on curated versions of these datasets. To evaluate the impact of curation, we trained MaskedMimic policies on both the curated and uncurated versions of these datasets. As shown in Table below, policies trained on the curated LAFAN1 and AMASS datasets consistently outperform those trained on the uncurated versions. This demonstrates that data curation improves policy performance despite the reduction in dataset size, validating that data quality is more critical than data quantity for learning robust motion tracking policies.
>
> | Dataset | PHUMA Test | | | | | Unseen Video | | | | |
> | :---: | :---: | :---: | :---: | :---: | :---: | :---: | :---: | :---: | :---: | :---: |
> | | Total | Stationary | Angular | Vertical | Horizontal | Total | Stationary | Angular | Vertical | Horizontal |
> | LaFAN1 (no curation) | 49.4 | 63.8 | 38.6 | 26.2 | 65.8 | 44.0 | 69.8 | 45.8 | 23.4 | 28.6 |
> | LaFAN1 | 56.8 | 76.0 | 45.7 | 30.0 | 68.7 | 49.6 | 84.5 | 52.2 | 23.4 | 26.4 |
> | AMASS (no curation) | 73.9 | 87.5 | 66.7 | 50.2 | **83.8** | 71.2 | 93.1 | 74.4 | 53.2 | **54.9** |
> | **AMASS** | **79.5** | **89.9** | **76.1** | **61.1** | 69.5 | **72.8** | **93.3** | **78.2** | **65.5** | 47.3 |

---

### Official Review · Reviewer_1eFX · 2025-10-31

**Soundness:** 2
**Presentation:** 2
**Contribution:** 2
**Rating:** 2
**Confidence:** 4

**Summary:**

The paper presents PHUMA, a large-scale physically grounded humanoid motion dataset designed for stable imitation learning in simulation. It addresses the problem that motion data extracted from Internet videos, such as Humanoid-X, often contain artifacts like floating, penetration, and joint-limit violations that degrade physics-based policy training. PHUMA introduces a physics-aware curation process that filters motions for contact and balance consistency, and a physics-constrained retargeting algorithm, PhySINK, that enforces non-floating, non-penetration, and non-skating constraints. The resulting dataset contains about 76k motion clips covering 73 hours of locomotion data. Experiments on the Unitree G1 and H1-2 humanoids in Isaac Gym show that policies trained on PHUMA achieve substantially higher imitation success and physical stability than those trained on LaFAN1, AMASS, or Humanoid-X.

**Strengths:**

1. The paper focuses on physically grounded humanoid motion, addressing a gap in large-scale imitation datasets where stability and contact consistency are often ignored.

2. The proposed PhySINK pipeline is technically solid and produces cleaner motion data with fewer artifacts than Humanoid-X.

3. The experiments are comprehensive within simulation, demonstrating clear quantitative improvements on multiple humanoid platforms and providing a useful dataset that can benefit future physics-based imitation learning research.

**Weaknesses:**

1. While the paper presents a clean and useful dataset, its novelty appears somewhat limited relative to recent works that also improve over Humanoid-X through physics-aware retargeting. Methods such as ASAP (RSS 2025), KunfuBot (NeurIPS 2025), and GMR have already introduced substantial innovations in motion retargeting and data cleaning—ASAP integrates RL-based physical simulation during retargeting, KunfuBot applies extensive filtering for realistic contacts, and GMR focuses specifically on retargeting fidelity—whereas PHUMA mainly improves data quality through curation and constraints. Compared with those works, the contribution here lies primarily in dataset refinement rather than methodological advance.

2. In addition, the dataset only covers locomotion, leaving out other interaction behaviors, and all results are limited to simulation without any hardware validation. There is no qualitative simulation videos for reference, which makes it difficult to assess the realism of the resulting motions.

**Questions:**

The imitation results are reported on both Unitree G1 and H1-2. Are the same PHUMA motions directly retargeted to each robot, or are robot-specific shape/scale adjustments applied?

Are there qualitative examples where PhySINK still fails?

---

> ### Author Response · Authors · 2025-11-27
>
> Hello Reviewer 1eFX,
>
> Thank you for your constructive feedback and for engaging closely with our work. We appreciate your acknowledgment of the dataset's utility and are happy to address your concerns regarding novelty, scope, and validation.
>
> > W1: While the paper presents a clean and useful dataset, its novelty appears somewhat limited relative to ~ ASAP (RSS 2025), KungfuBot (NeurIPS 2025), and GMR~ Compared with those works, the contribution here lies primarily in dataset refinement rather than methodological advance.
>
> We appreciate your thoughtful assessment. While dataset curation is an important component of our work, we believe our contribution extends beyond this to address a critical gap in existing humanoid motion datasets.
>
> Existing datasets face a fundamental tradeoff: they are either physically reliable but limited in scale (e.g. LaFAN1, LocoMuJoCo), or large-scale but lack physical reliability (e.g. Humanoid-X). Our work bridges this gap by providing both: (1) a large-scale locomotion dataset with physical reliability, and (2) a scalable pipeline (PhySINK) that can efficiently generate physically reliable humanoid motions from diverse human motion sources.
> Beyond scale, our approach differs from prior work in several key aspects:
>
> 1) The ASAP methodology achieves clean motions by filtering out motions that a imitation policy fails to track in a physics simulator. While this approach yields clean data, the resulting dataset is potentially biased since the filtering process is constrained by the tracking performance of the imitation policy. In contrast, PHUMA's filtering is based purely on physics-based heuristics, allowing us to retain a more diverse  motion distribution that subsequently enables the training of more generalizable policies.
>
> 2) The KungfuBot method contributes foot contact information by using a simple binary threshold on foot height to define contact. On the other hand, our graded contact information is estimated during the human motion curation phase. We then leverage this contact information in the ground loss component of our PhySINK retargeting method. This precise use of contact leads to retargeted motions with more human-like foot behavior, which is often neglected.
>
> 3) We fully agree that comparing with GMR is essential. We have included a comprehensive comparison between GMR and PhySINK in the general response #2. Please refer to the general response #2 for detailed explanations and quantitative results.
>
> > W2: In addition, the dataset only covers locomotion, leaving out other interaction behaviors, and all results are limited to simulation without any hardware validation. There is no qualitative simulation videos for reference, which makes it difficult to assess the realism of the resulting motions.
>
>
> While the main focus of our work is constructing a physically plausible humanoid dataset sourced from diverse human data, we fully agree that sim-to-real deployment would be valuable for (1) stronger validation of our data pipeline and (2) assessing the realism of the resulting motions.
>
> (1) Given that we currently lack hardware access, we newly conducted sim-to-sim (IsaacGym-to-MuJoCo) experiments. The results in MuJoCo show that policies trained on PHUMA achieve superior performance compared to those trained on AMASS, which is consistent with our results in IsaacGym.
>
> (2) To assess the quality of our data, we have prepared qualitative videos in the link ([anonymous.video_link](https://anonymous-robotics-researcher.github.io/Paper_10482/)), including: diverse sample motions in PHUMA, comparisons of PhySINK against other retargeting methods (including GMR), and motion tracking visualizations (including sim-to-sim experiments), and failure cases of PhySINK.

---

> ### Author Response · Authors · 2025-11-27
>
> > Q1: The imitation results are reported on both Unitree G1 and H1-2. Are the same PHUMA motions directly retargeted to each robot, or are robot-specific shape/scale adjustments applied?
>
> Before retargeting, we apply a robot-specific body shape optimization using the T-pose, the method used in H2O[1]. This results in a customized, robot-specific shape that is then used for the PhySINK retargeting. This contrasts with GMR, which relies on a hard scaling factor derived from the SMPL $\beta$ component.
>
> [1] He et al, "Learning Human-to-Humanoid Real-Time Whole-Body Teleoperation", IROS 2024.
>
> > Q2: Are there qualitative examples where PhySINK still fails?
>
> While PhySINK successfully retargets most human motions into physically reliable robot motions, there are limitations in certain scenarios. PhySINK currently fails to handle motions with: (1) significant vertical root translation (e.g., climbing stairs or ladders), and (2) ground contact involving body parts other than the feet (e.g., upper body contact during floor-based movements). We provide video examples of these failure cases at the bottom of the link  ([anonymous.video_link](https://anonymous-robotics-researcher.github.io/Paper_10482/)).
>
> Our current work focuses primarily on bipedal locomotion where the feet serve as the primary ground contact points (e.g., walking, running, jumping). Extending PhySINK to handle object interactions (e.g., climbing) and multi-contact movements involving knees, hands, or upper body is an interesting direction for future work.

---

### Official Review · Reviewer_h4PJ · 2025-11-01

**Soundness:** 3
**Presentation:** 3
**Contribution:** 3
**Rating:** 8
**Confidence:** 3

**Summary:**

PHUMA introduces a new dataset of physically-grounded humanoid motions reconstructed then processed from videos of humans. The pipeline extracts human motion from RGB video then retargets it using a proposed Physically-grounded Shape-adaptive Inverse Kinematics (PhySINK) procedure. For policies trained on the PHUMA dataset, the authors observe higher success rates than corresponding policies trained on AMASS and Humanoid-X.

**Strengths:**

- Data curation and processing pipeline is well-justified and thoroughly explained.
- PhySINK has good quantitative comparison to SINK and Mink's IK as appropriate baselines.
- Method identifies real problems with existing video-to-humanoid motion pipelines that lead to poor downstream performance during RL training and deployment.
- The dataset is relatively large, comparing favorably against popular datasets like AMASS
- Data pipeline validated on multiple humanoid robot form factors.

**Weaknesses:**

- 1.2x success rate improvement over AMASS isn't very dramatic.
- "Physically grounded" in this paper means the motions have kinematic plausibility as defined by heuristics and loosely verified by downstream sim RL training. I think this is somewhat misleading wording, since I would interpret "physically grounded" to mean that it is simulated dynamically in the data processing loop.
- A sim-to-real deployment would be a stronger validation of the data pipeline.

**Questions:**

- Why is VIBE used as opposed to a newer model like VIMO?

---

> ### Comment · Reviewer_h4PJ · 2025-11-27
>
> I am lowering my score due to lack of author response and some new concerns. I have found the retargeting method is less novel than I had originally thought and missing proper citation. I'm lowering my score to a 4.

---

> > ### Author Response · Authors · 2025-11-27
> >
> > Hello Reviewer h4PJ.
> > Thank you again for your thoughtful review and insightful comments on our work. Your feedback is highly valuable, and we appreciate the opportunity to clarify and address your concerns.
> >
> > Here are our detailed responses to your specific points:
> >
> > > W1: 1.2x success rate improvement over AMASS isn't very dramatic.
> >
> > We thank the reviewer for the insightful comment. We agree that the quantitative improvement is not dramatic. However, we believe our data pipeline enables the scalable collection of physically reliable motions from video sources, whereas datasets like AMASS are constrained by the finite volume of motion capture data.
> >
> > > W2: "Physically grounded" in this paper means the motions have kinematic plausibility as defined by heuristics and loosely verified by downstream sim RL training. I think this is somewhat misleading wording, since I would interpret "physically grounded" to mean that it is simulated dynamically in the data processing loop.
> >
> > We thank the reviewer for the insightful comment regarding the terminology. We agree that "physically grounded" could be misleading. To address this, we have revised the terminology throughout the manuscript: we now refer to the dataset as "physically plausible" and the retargeting method (PhySINK) as "physically constrained."
> >
> >
> > > W3: A sim-to-real deployment would be a stronger validation of the data pipeline.
> >
> > We fully agree that a sim-to-real deployment offers the strongest validation for a robotics data pipeline. As detailed in our common response, current circumstantial limitations prevent us from deploying the policy on real-world hardware.
> >
> > To provide the best possible alternative validation, we have prepared an extensive sim-to-sim performance evaluation, which includes both quantitative metrics and qualitative demonstrations (as seen in the Sim-to-Sim performance section on the Experiment part). This validation utilizes an established sim-to-real benchmark (KungfuBot2) to maximize the confidence in our data's transferability.
> >
> > > Q1: Why is VIBE used as opposed to a newer model like VIMO?
> >
> > For most of the curated motions in PHUMA derived from Humanoid-X, the original videos are not publicly released. Consequently, for those motions, we start from existing reconstructed human motions. However, for our self-collected video dataset “PHUMA Video”, we utilize TRAM (which incorporates VIMO), for reconstructing human motion. Please refer to Appendix C.1 for more details.

---

### Author Response · Authors · 2025-11-27
**General Response**

We sincerely thank all reviewers for your thorough, constructive, and insightful feedback. We apologize for the delayed response and truly appreciate the time and effort you dedicated to reviewing our work. We've identified several commonly raised points across the reviews, and we address them below:


> 1. Need for Video Demonstrations

We have prepared comprehensive video demonstrations and qualitative comparisons on our supplementary website to provide a clearer visualization of our results. The videos, accessible via the following link ([anonymous.video_link](https://anonymous-robotics-researcher.github.io/Paper_10482/)), include:
- Example of the PHUMA Dataset
- Qualitative Results of Existing Retargeting Methods:
    - IK, SINK, and PhySINK
    - Comparison between GMR and PhySINK
- Imitation Performance of the MaskedMimic Policy
- Path Following Performance
- Imitation Performance of the KungfuBot Policy
- Sim-to-Sim Performance Comparison


> 2. Comparison with Physically Grounded Retargeting Method (GMR)


Both GMR and PhySINK follow a two-stage retargeting pipeline with different approaches to generating target motions.

GMR employs a heuristic scaling strategy: it estimates the source subject's height from the first SMPL shape parameter ($\beta_0$), computes a scaling ratio $r = H_{src} / H_{ref}$ relative to a reference height, and applies this to limb-specific factors to generate target keypoints. This approach is effective for average-height subjects. However, we observe that linear scaling may face challenges when accommodating the nonlinear relationship between human morphology and robot kinematic constraints. For subjects with significant height variations, this can lead to artifacts—for example, unreachable targets for tall subjects (e.g., 1.5m motions for a 1.3m Unitree G1) or compressed targets requiring deep crouches for shorter subjects (see supplementary page).

PhySINK addresses these challenges through shape optimization. First, it optimizes the SMPL shape parameters ($\beta$) to align human and humanoid proportions in a shared T-pose, then applies these parameters to the original motion sequence while preserving joint angles. Second, it optimizes the humanoid's joint angles using physics-based constraint losses (Section 3.2). This approach aims to maintain kinematic feasibility and physical plausibility across diverse subject morphologies.

We compare both methods in Table 2 (retargeting quality)
| | Motion Fidelity (%) | Joint Feasibility (%) | Non-Floating (%) | Non-Penetration (%) | Non-Skating (%) |
| :--- | :---: | :---: | :---: | :---: | :---: |
| IK | 27.6 | 91.7 | 55.6 | 47.8 | 59.7 |
| GMR | 56.3 | 81.8 | 14.7 | 100.0 | 67.7 |
| SINK | 94.8 | 95.9 | 96.4 | 14.9 | 55.4 |
| + Joint Feasibility Loss | **94.9** | **100.0** | 96.4 | 14.8 | 55.6 |
| + Grounding Loss | **94.9** | **100.0** | **99.9** | **97.2** | 53.6 |
| + Skating Loss = **PhySINK** | 94.8 | **100.0** | **99.9** | 96.8 | **89.7** |

 and Table 9 (policy performance).

| Dataset | **PHUMA Test** | | | | | **Unseen Video** | | | | |
| :--- | :---: | :---: | :---: | :---: | :---: | :---: | :---: | :---: | :---: | :---: |
| | Total | Stationary | Angular | Vertical | Horizontal | Total | Stationary | Angular | Vertical | Horizontal |
| GMR | 84.0 | 92.1 | 77.8 | 77.1 | 89.1 | 75.2 | **99.1** | 77.8 | **61.7** | 52.7 |
| PhySINK | **89.9** | **94.2** | **87.6** | **84.2** | **91.8** | **81.7** | 97.4 | **86.7** | **61.7** | **71.4** |

 Our results suggest that PhySINK's shape optimization approach offers improvements in motion fidelity preservation across varying body types. Please see Section 3.2 and Appendix A.3 for additional details.

---

> ### Author Response · Authors · 2025-11-27
>
> > 3. Contribution of Physics-Based Filtering
>
>
> To determine the impact of physics-based filtering components on motion tracking policy learning, we conducted ablation studies using the MaskedMimic policy. We generated several datasets by selectively applying filtering components to the base Humanoid-X dataset, followed by PhySINK retargeting. We excluded the pre-retargeted LaFAN1 and LocoMuJoCo datasets to isolate the effect of filtering quality. The datasets are:
>
> - Humanoid-X (No filtering)
> - Humanoid-X + Jerk Filtering
> - Humanoid-X + Foot Contact Filtering
> - Humanoid-X + Height Filtering
> - Humanoid-X + Base of Support Filtering
> - Humanoid-X + All filtering applied
>
> The retargeting quality of each dataset is shown in Table 19.
>
> | Motion Source | Hours | Motion Fidelity (%) | Joint Feasibility (%) | Non-Floating (%) | Non-Penetration (%) | Non-Skating (%) |
> | :--- | :---: | :---: | :---: | :---: | :---: | :---: |
> | Humanoid-X | 237.2 | 70.6 | **100.0** | 98.7 | 92.2 | 90.1 |
> | Jerk Filter | 141.1 | 76.9 | **100.0** | 99.7 | 96.5 | **92.0** |
> | Foot Contact Filter | 123.1 | 77.1 | **100.0** | 99.8 | **96.8** | 90.6 |
> | Height Filter | 135.7 | 87.0 | **100.0** | 99.4 | 94.7 | 89.8 |
> | BoS Filter | 110.3 | 90.7 | **100.0** | 99.4 | 94.9 | 90.0 |
> | All Filter | 62.2 | **94.8** | **100.0** | **99.9** | 96.7 | 89.7 |
>
> Our results in Table 20
>
> | Motion Source | **PHUMA Test** | | | | | **Unseen Video** | | | | |
> | :--- | :---: | :---: | :---: | :---: | :---: | :---: | :---: | :---: | :---: | :---: |
> | | Total | Stationary | Angular | Vertical | Horizontal | Total | Stationary | Angular | Vertical | Horizontal |
> | Humanoid-X | 85.7 | 92.6 | 82.2 | 75.5 | 89.5 | 75.4 | 96.6 | 81.3 | 58.5 | 52.7 |
> | Jerk Filter | 87.1 | 93.1 | 83.2 | 80.4 | 90.9 | 78.8 | **100.0** | 76.8 | **64.9** | 70.3 |
> | Foot Contact Filter | 88.2 | 93.9 | 84.9 | 81.8 | 90.5 | 78.6 | 95.7 | 82.3 | 61.7 | 65.9 |
> | Height Filter | 86.9 | 93.0 | 83.2 | 81.6 | 88.9 | 77.2 | 94.8 | 79.8 | 59.6 | 67.0 |
> | BoS Filter | 86.8 | 92.7 | 83.6 | 79.1 | 89.8 | 77.0 | 96.6 | 79.8 | 61.7 | 61.5 |
> | ALL Filter | **89.9** | **94.2** | **87.6** | **84.2** | **91.8** | **81.7** | 97.4 | **86.7** | 61.7 | **71.4** |
>
>  indicate that datasets with at least one physics-based filtering component consistently outperform the unfiltered baseline. Applying all filters yields the best performance, despite the resulting dataset being approximately 4× smaller than the unfiltered version and 2× smaller than datasets with individual filters. This demonstrates that data quality is more critical than data quantity for learning robust motion tracking policies. See Appendix D.2 for detailed explanations.

---

> ### Author Response · Authors · 2025-11-27
>
> > 4. Effect of Physical Artifacts on Humanoid Motion Imitation Learning
>
> To investigate how physical artifacts introduced during motion retargeting affect policy learning, we trained MaskedMimic policies on datasets derived from the same human motion sources but retargeted using different methods:
>
> - IK
> - GMR (optimized version of IK)
> - SINK (Baseline with artifacts)
> - SINK + Joint Feasibility Loss
> - SINK + Joint Feasibility Loss, Grounding Loss
> - PhySINK (Minimally-artifacted data)
>
> The quantitative retargeting quality metrics are shown in Table 2.
>
> | | Motion Fidelity (%) | Joint Feasibility (%) | Non-Floating (%) | Non-Penetration (%) | Non-Skating (%) |
> | :--- | :---: | :---: | :---: | :---: | :---: |
> | IK | 27.6 | 91.7 | 55.6 | 47.8 | 59.7 |
> | GMR | 56.3 | 81.8 | 14.7 | 100.0 | 67.7 |
> | SINK | 94.8 | 95.9 | 96.4 | 14.9 | 55.4 |
> | + Joint Feasibility Loss | **94.9** | **100.0** | 96.4 | 14.8 | 55.6 |
> | + Grounding Loss | **94.9** | **100.0** | **99.9** | **97.2** | 53.6 |
> | + Skating Loss = **PhySINK** | 94.8 | **100.0** | **99.9** | 96.8 | **89.7** |
>
> As shown in Table 21,
>
> | Dataset | **PHUMA Test** | | | | | **Unseen Video** | | | | |
> | :--- | :---: | :---: | :---: | :---: | :---: | :---: | :---: | :---: | :---: | :---: |
> | | Total | Stationary | Angular | Vertical | Horizontal | Total | Stationary | Angular | Vertical | Horizontal |
> | IK | 70.5 | 85.9 | 63.3 | 47.6 | 77.3 | 68.5 | 96.6 | 72.4 | 43.6 | 49.5 |
> | GMR | 84.0 | 92.1 | 77.8 | 77.1 | 89.1 | 75.2 | 99.1 | 77.8 | 61.7 | 52.7 |
> | SINK | 89.1 | 94.0 | 86.0 | 84.9 | 90.7 | 79.0 | **100.0** | 81.8 | **62.8** | 62.6 |
> | + Joint Feasibility Loss | 87.0 | 92.1 | 83.6 | 79.8 | 90.9 | 78.6 | 94.8 | 84.7 | 56.4 | 67.0 |
> | + Grounding Loss | **90.0** | 93.5 | **87.9** | **85.6** | **92.0** | 80.4 | 98.3 | 86.2 | 60.6 | 64.8 |
> | + Skating Loss= **PhySINK**| 89.9 | **94.2** | 87.6 | 84.2 | 91.8 | **81.7** | 97.4 | **86.7** | 61.7 | **71.4** |
>
> policies trained on SINK-based methods consistently outperform those trained on IK-based retargeting. Notably, artifacts such as floating, penetration, and skating have minimal impact on policy performance. Instead, motion fidelity—the degree to which retargeted motion preserves the original kinematics—emerges as the most critical factor.
>
> SINK-based methods achieve superior motion fidelity compared to IK-based approaches due to fundamental differences in their retargeting strategies. IK-based methods apply heuristic scaling to bridge morphological differences between humans and humanoids, which can distort the original motion. In contrast, SINK-based methods first optimize the humanoid body shape to match human proportions, then apply the original joint angles directly, thereby preserving the kinematic structure of the source motion.
> Please refer to Appendix D.3 for detailed explanations.

---

> > ### Author Response · Authors · 2025-12-01
> >
> > > 5. Lack of Hardware Deployment Results
> >
> > We fully agree that sim-to-real validation is critical for robotics research. However, we currently lack access to the target hardware platform required for real-world deployment. To address this limitation, we conducted a comprehensive sim-to-sim validation experiment using the well-established KungfuBot2 [1] policy, which has demonstrated strong sim-to-real transfer capabilities.
> >
> > Our validation methodology proceeded as follows: We trained an MLP policy in the Isaac Gym simulator using both our PHUMA dataset and the widely used AMASS dataset. We then performed zero-shot evaluation in the MuJoCo simulator on unseen motions, providing evidence of cross-simulator generalization—a key indicator of sim-to-real transferability.
> >
> > Performance in Isaac Gym (Training Simulator):
> > | Dataset | PHUMA Test | | | | | Unseen Video | | | | |
> > | :---: | :---: | :---: | :---: | :---: | :---: | :---: | :---: | :---: | :---: | :---: |
> > | | Total | Stationary | Angular | Vertical | Horizontal | Total | Stationary | Angular | Vertical | Horizontal |
> > | AMASS | 66.6 | 86.7 | 58.8 | 41.8 | 68.4 | 67.7 | **97.4** | 68.0 | 59.6 | 37.4 |
> > | **PHUMA** | **82.9** | **93.5** | **78.5** | **74.2** | **81.7** | **73.8** | **97.4** | **76.9** | **63.8** | **47.3** |
> >
> > Performance in MuJoCo (Evaluation Simulator):
> > | Dataset | PHUMA Test | | | | | Unseen Video | | | | |
> > | :---: | :---: | :---: | :---: | :---: | :---: | :---: | :---: | :---: | :---: | :---: |
> > | | Total | Stationary | Angular | Vertical | Horizontal | Total | Stationary | Angular | Vertical | Horizontal |
> > | AMASS | 62.1 | 81.4 | 54.2 | 38.8 | 64.3 | 64.3 | 86.2 | 68.5 | 54.3 | 37.4 |
> > | **PHUMA** | **75.0** | **87.6** | **69.3** | **61.6** | **76.3** | **70.0** | **87.9** | **78.8** | **59.6** | **38.5** |
> >
> > The results demonstrate that the PHUMA-trained policy consistently outperforms the AMASS-trained policy across all motion categories in both simulators. Notably, this performance advantage transfers from the training simulator (Isaac Gym) to the evaluation simulator (MuJoCo), indicating that PHUMA's superior data size and quality contribute to better cross-simulator generalization. For detailed analysis, please refer to Section 4.5.
> >
> > [1] Han et al, "KungfuBot 2: Learning Versatile Motion Skills for Humanoid Whole-Body Control", arXiv 2025.

---

### Meta-Review · Area_Chair_1tQH · 2026-01-07

**Summary:**

The paper got mixed ratings from the reviewers (6, 2, 8, and 4). The reviewer who initially gave a rating of 8 indicates an intention to reduce the score to 4, prior to considering the authors’ responses.

Reviewers commonly value the advantages of the dataset curated in this work, particularly the proposed filtering and optimization procedures. They also agree that the dataset demonstrates effectiveness within an imitation learning framework.

However, reviewers raise significant concerns regarding the novelty of the proposed methods, noting that similar physics-aware retargeting approaches are already presented in prior work, such as ASAP, KungFuBot, and GMR. In addition, reviewers point out the absence of comparisons with GMR, missing methodological details, the limited diversity of the dataset, which focuses primarily on locomotion, and the lack of real-world demonstrations.

**Reviewer Concerns:**

The authors’ rebuttal partly addresses the reviewers’ concerns. The authors provide qualitative videos, which are missing in the original submission, and add comparisons with GMR. The authors’ responses also include sim-to-sim experiments to partially approximate real-world testing, and also provide additional ablations.

**Reviewer Scores:**

The AC carefully reviews the paper, the reviewers’ original comments, the authors’ responses, and the response summary. Overall, the AC believes that the reviewers’ scores are likely to remain unchanged or potentially decrease.

The AC largely agrees with the reviewers’ major concerns regarding the novelty of the paper. While the effectiveness of the proposed filtering and optimization pipeline is clear, most components are based on commonly used modeling constraints, such as enforcing joint feasibility, preventing interpenetration, and avoiding foot sliding. Since the method explicitly reduces these artifacts, it is expected that such errors decrease relative to competing approaches in these metrics (e.g., Table 2)

The AC finds that the central message of the paper is that filtering and optimizing motion samples (into physically plausible forms) improves imitation learning performance. However, this conclusion is already emphasized in prior work, particularly GMR, which demonstrates that retargeting quality significantly impacts downstream imitation learning. As a result, the paper’s contributions and experimental findings appear to be an extension of existing work (with further filtering and further objective terms for the optimization) rather than introducing a clearly distinct or novel direction.

Based on these considerations, the AC concludes that the current version of the paper does not meet the acceptance threshold.

---

### Decision · Program_Chairs · 2026-01-26

Reject